# Developing indicators of risk to environmental variability based on species dependency in U.S. fishing communities in the Northeast and Southeast Regions

**Tarsila Seara** [1]*, **Matthew McPherson**[2], **Patricia M. Clay** [3], **Michael Jepson**[2], **Lisa L. Colburn**[4], **Changhua Weng**[5], **Angela Silva**[6]

**1** NOAA NMFS Northeast Fisheries Science Center, Milford Laboratory, Milford, Connecticut, United States of America, **2** NOAA NMFS Southeast Fisheries Science Center, Miami, Florida, United States of America, **3** NOAA NMFS Northeast Fisheries Science Center, Woods Hole, Massachusetts, United States of America, **4** NOAA NMFS, Office of Science and Technology, Silver Spring, Maryland, United States of America, **5** ECS Federal, in support of Social and Economic Analysis Division, Office of Science and Technology, NOAA Fisheries, Silver Spring, Maryland, United States of America, **6** NOAA NMFS Northeast Fisheries Science Center, Narragansett Laboratory, Narragansett, Rhode Island, United States of America

\* tarsila.seara@noaa.gov

## Abstract

Fishing communities worldwide have or are likely to experience social, economic, and cultural impacts from environmental variability. Changes in marine fisheries will require adaptation by fishing communities and fisheries managers alike. Here, Community Environmental Variability Risk Indicators (CEVRI) were developed to assess risk to environmental change for fishing communities in the U.S. Northeast and Southeast Regions based on spatial and temporal trends between 2000 and 2022. To accomplish this, we analyzed commercial landings value as it relates to species level Climate Vulnerability Assessment (CVA) scores for species considered commercially, recreationally, and ecologically important. The CVA considers the vulnerability of species to 12 sensitivity and 12 exposure factors relating to important environmental factors within the regional context. Here, we used three sensitivity factors: Stock Size/Status, Ocean Acidification, and Temperature, as well as Total Sensitivity and Total Vulnerability. Community level scores were used to analyze intra and inter region variation, and to understand trends in community risk as revenue dependence on different species changes through time. In general, communities in the Gulf of America/Florida Keys sub-region presented lower risk to the factors analyzed than the South Atlantic sub-region and the Northeast. Ocean Acidification was the sensitivity factor with the highest levels of risk for communities. The findings of this study have important applications to inform decision-making and to help communicate environmental variability associated risks to broader audiences, thus further developing the ability of stakeholders to understand and assess cumulative impacts and complex trade-offs affecting the sustainability of marine ecosystems and resources.

**Data availability statement:** All relevant data are within the manuscript and its Supporting information files.

**Funding:** Funding for this project was provided by the NOAA Fisheries Office of Science and Technology under the Inflation Reduction Act (IRA).

**Competing interests:** The authors have declared that no competing interests exist.

## Introduction

Environmental variability significantly affects marine ecosystems in both expected and unpredictable ways. Uncertainty driven by environmental variability needs to be accounted for in management processes, and this need is particularly prominent in ecosystem-based approaches [1]. Current and projected primary effects of environmental variability affecting marine ecosystems and coastal communities include fluctuations in stock population due to habitat degradation, hurricanes, sea level rise, ocean temperature changes, and ocean acidification [2,3]. Changes in the physical and chemical composition of the oceans result in substantial direct and indirect impacts on marine life. These impacts may lead to significant changes in species richness and diversity stemming from mass mortality and population distribution shifts due to changed ecosystem conditions [4–6].

Coastal fishing communities are greatly affected by environmental variability, both because of their physical location and because of their frequent social, cultural, and economic dependence on fishing. These impacts are expected to become more pressing as global environmental changes become more extensive [7–10]. Changes in ocean temperature and acidification affecting marine life have the potential to directly impact fisheries and fishery-dependent livelihoods [11–15]. Coastal fishing communities worldwide have or are likely to experience social, economic, and cultural impacts from environmental variability, both negative (e.g., loss of infrastructure, fish stock decline) and positive (e.g., increased abundance of certain valuable species) [16,17]. Changes in marine fisheries as a consequence of environmental variability will require adaptation by coastal fishing communities and fisheries managers alike [18,19].

In this study, we examine community level risk to environmental variability in coastal fishing communities in the Northeast and Southeast U.S. using indicators based on species dependency and their respective bioenvironmental vulnerabilities as assessed by regional experts. Because of sub-regional ecosystem differences in the Southeast, we subdivide it into two sub-regions: the South Atlantic and the Gulf of America (formerly Gulf of Mexico)/Florida Keys (Fig 1). These sub-regions reflect the structure used by the Fishery Management Councils overseeing fisheries management in the region.

### Impacts of environmental variability on fisheries

In the first two decades of the 21ˢᵗ century (2001–2020), the average global surface temperature was 0.99°C higher than in the period between 1850 and 1900 and 1.09°C higher between 2011 and 2020 alone than the same historical baseline (1850–1900) [21]. IPCC (ibid: 14) best estimates predict an average global surface air temperature increase of 1.4 to 4.4°C by 2100. Best estimates of ocean warming by the end of the 21st century are approximately 0.6°C to 2.0°C in the top 100 meters, and about 0.3°C to 0.6°C at a depth of about 1 km [22]. Similarly, [23] report increases in ocean temperature overall, but especially in the upper 700 meters. Exposure of marine species to increases in temperature can have a direct effect on their physiology [24,25] and life cycle events [26,27]. Changes in temperature and

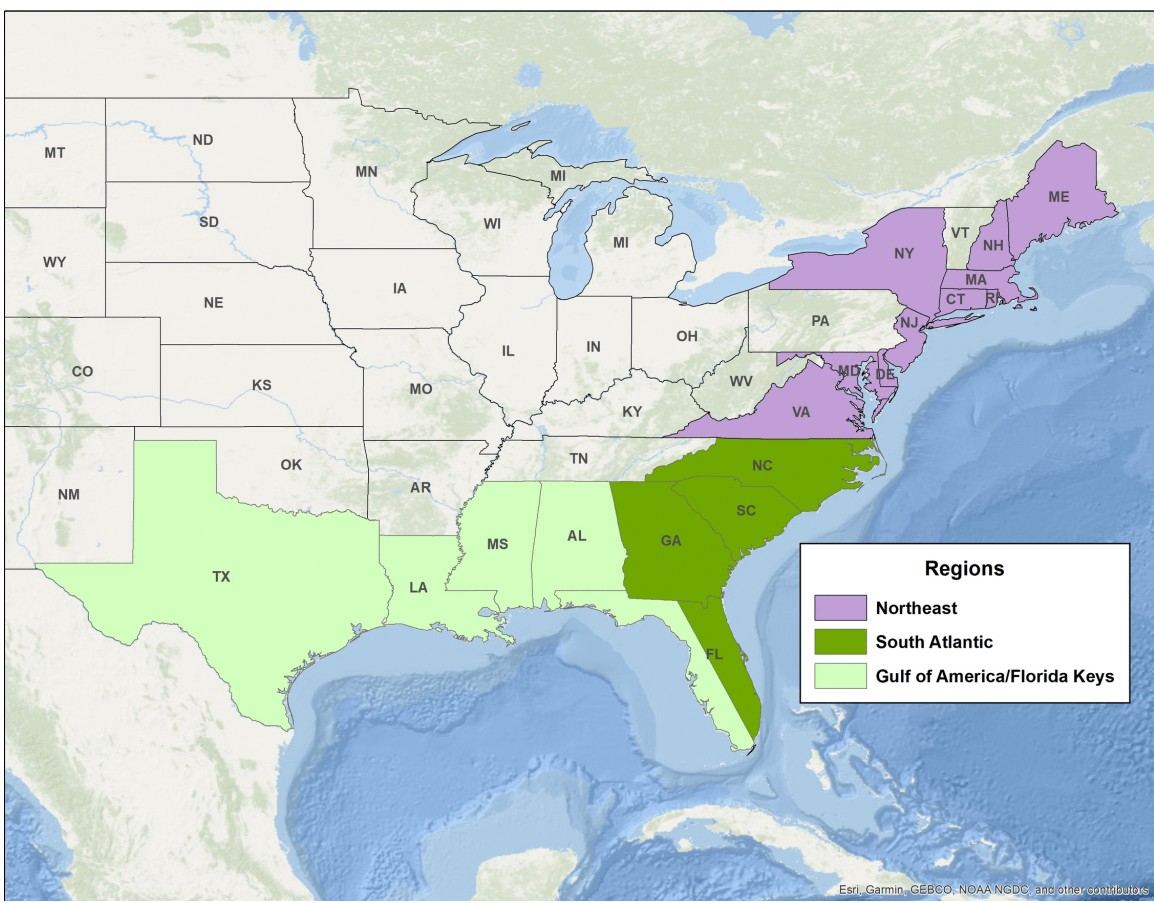

**Fig 1. Map of the study area showing region and sub-region boundaries.** The Northeast Region includes the states from Maine through Virginia. The South Atlantic sub-region of the Southeast includes the states from North Carolina through Georgia, plus the east coast of Florida. The Gulf of America/Florida Keys sub-region includes the Florida Keys, the west coast of Florida and the states from Alabama through Texas. Data source: [20]. The basemap is provided by Esri, GEBCO, NOAA, National Geographic, and other contributors.

habitat have also been shown to affect the distribution of marine populations, as organisms seek optimal habitat conditions to increase their overall survival [28–33]. Other studies have shown that both increasing interannual variability in continental shelf temperatures and a general warming trend have led to poleward shifts in many marine species [34–39].

Ocean acidification is also an increasing threat to marine life. The reaction of carbon dioxide with seawater forms weak carbonic acid which reduces seawater pH, increasing the acidity of the ocean [40]. Ocean acidification is predicted to continue to increase throughout the 21$^{st}$ century [21]. Increased acidity has been shown to have negative impacts on the reproductive performance of marine fish [41], fish behavior [42], habitat [43–45], and the resilience of food webs [46,47]. Studies show that the depletion of carbonate in seawater due to increased acidification makes it difficult for organisms that build and maintain calcium carbonate shells to carry out normal life cycles, affecting adult, juvenile, and larval stages (see [48]).

The Atlantic Ocean off the Northeast Region (U.S. Northeast Continental Shelf Large Marine Ecosystem (NES LME)) has experienced increases in both sea surface temperatures and ocean acidification. In some projections, sea surface temperatures may warm about 2.5 degrees Celsius and the pH might reach critical levels by 2050 for the entire Gulf of

Maine [49]. Over the past 100 years, sea surface temperatures in the NES LME have warmed at a higher rate than in any other NOAA Fisheries management region [50]. Studies have shown significant northward shifts as well as reduced productivity in NES LME species with commercial and recreational importance, including Atlantic cod (*Gadus morhua*), sea scallops (*Placopecten magellanicus*), and American lobster (*Homarus americanus*), as a consequence of increased sea surface temperatures and ocean acidification [35,51–57].

The effects of a changing environment have also been observed in the U.S. South Atlantic and the Gulf of America/ Florida Keys. [58] noted changes in several environmental variables in the South Atlantic including increasing annual and decadal sea surface temperature, and accelerated rates of sea level rise since 2010. While information on species distribution shifts for the Southeast suggests less pronounced trends when compared to the Northeast region, NOAA Fisheries assessments [59] show significant northward shifts for select species since the 1980s. [60] point to the exploitation of blueline tilefish (*Caulolatilus microps*, Malacanthidae) for nearly a decade before management adapted to its northward expansion from the South Atlantic to Mid-Atlantic. [61] identify shifts in distribution of black sea bass in the South Atlantic region and [62] note that during workshops fishers reported changes in distribution and abundance for species such as dolphinfish (*Coryphaena hippurus*, Coryphaenidae) and hypothesized temperature changes as a cause. [63] also show potential effects of a changing environment on the distribution of several marine species in the Region. In addition, increasingly intense environmental shifts have led to numerous fishery disasters in all regions but more severely in the Southeast [64] -- primarily from hurricanes, but also increased freshwater flooding [65] and algal blooms [66], both of which have affected certain vulnerable species.

Many coastal fishing communities predicted to be at risk to significant variation in environmental factors are already facing impacts from overfishing, pollution, and coastal zone modifications [67,68] and may possess low adaptive capacity as a result [69–71]. Changes in stock abundance and the availability of fish products can negatively affect total revenue and fishing costs as well as individual and community well-being [14,70,72,73]. Environmental variability risk for fishing and coastal communities are determined by 1) their exposure to variability; 2) their sensitivity to change in terms of target species and the ecosystem on which they depend; and 3) their ability to adapt to change (i.e., their adaptive capacity) [74–77].

In this study, we focus on the second factor described above by using an interdisciplinary approach to develop indicators that assess U.S. Northeast and Southeast Regions coastal fishing communities' risk to environmental variability and analyze spatial and temporal trends and changes occurring between 2000 and 2022. To characterize fishing communities according to their environmental variability risk, we analyzed commercial landings value composition as it relates to species level Climate Vulnerability Assessment (CVA) scores first developed by [55] for species considered commercially, recreationally, and ecologically important in the Northeast (See [78] and [53] for the detailed methodology and analysis for the Northeast, [79] for the South Atlantic, and [80] for the Gulf of America/Florida Keys). While a similar methodology has been previously used to score communities based on overall contribution to landings value of species grouped by their vulnerability category (low to very high) [81], in this study we use a new method to analyze community risk. The new methodology is based on individual species' CVA scores for Total Sensitivity and Total Vulnerability, as well as three separate sensitivity attributes related to environmental variability: Temperature, Ocean Acidification, and Stock Size/Status. Another novelty of this study is the use of the indicators' scores to generate temporal trends, allowing for the visualization of community-level risk through time. The ability to operationalize risk to environmental variability at the coastal fishing community level will help to guide and inform the development of fisheries policy and management strategies that address impacts affecting both the sustainability of fisheries resources and coastal fishing communities' well-being in the face of change and uncertainty. Moreover, ecosystem-based management requires assessment of the interactions of social and environmental factors to understand the range of impacts from any ecosystem change, the management or policy change in response to those impacts, and the adaptations available to users of the resource. Therefore, adaptation strategies must be informed by both social and natural sciences in order to provide alternatives that meet ecosystem resilience and sustainability objectives as well as provide viable adaptations for fishers and their communities [82].

## Materials and methods

### Developing and calculating the Community Environmental Variability Risk Indicators

**Base data – Fisheries landings.** The data used for the development of the Community Environmental Variability Risk Indicators (CEVRI) consists of landed value reported to NOAA Fisheries by dealers and aggregated at the community level by port of landing for the Northeast region and dealer address for the Southeast region. The use of different aggregation criteria for the two regions is a consequence of limitations in the data reported by the responsible authorities. For the purposes of our analyses, communities are geographically defined by the U.S. Census Bureau Federal Information Processing Standard (FIPS) "Census Place" codes associated with a port or dealer's address city. The use of landings value as reported by dealers by location was used to reflect value likely to be retained in the community, thus indicating their revenue dependency on different species.

The contribution to total annual value landed in each community from 2000 to 2022 only for species classified by the CVA process was used in the indicators' calculation. The species' landed value in dollars was used, instead of weight in pounds, to more adequately relate risk to environmental variability to community economic dependence on vulnerable species, i.e., to assess potential revenue loss associated with fluctuations in the availability of target species due to certain environmental disturbances and conditions.

**Base data – Species Climate Vulnerability Assessment.** As noted above, the species CVA scores used in this study were developed by NOAA Fisheries scientists as part of a national effort conducted prior to and independently of the analyses presented in this study to classify species by climate and environmental vulnerability, using a methodology detailed in [78] (see also [59]). This methodology relies on expert knowledge and uses species profiles and scientific literature to score species for different sensitivity attributes and exposure factors. Sensitivity attributes refer to the biological characteristics of the species that are indicative of their ability/inability to respond to potential environmental changes. Exposure factors are defined as the overlap between the species' geographic distribution and the magnitude of expected environmental and habitat changes. As a final product of the CVA process, each species receives a score ranging from 1 (low) to 4 (very high) for each sensitivity and exposure factor (see Supplemental Material I for a complete list of sensitivity attributes and exposure factors). These scores are used to calculate Total Sensitivity (considering all sensitivity attributes) and Total Vulnerability (considering all sensitivity and exposure factors), generating two additional scores for each individual species. Species scores for both sensitivity attributes and exposure factors are region-specific and, while all sensitivity attributes and 6 exposure factors are common between all, some vary by region (see [78–80] for more details). The three separate sensitivity attributes used in this study, Stock Size/Status, Ocean Acidification, and Temperature are defined in Table 1. A total of 71 species have been CVA-classified for the South Atlantic [79], 75 for the Gulf of America/Florida Keys [80], and 82 for the Northeast Region [55].

**Community Level Indicators – Environmental Variability Risk.** The methodology in [83] was used to calculate the CEVRI for Northeast and Southeast Region coastal fishing communities. The indicators are calculated by multiplying the percent contribution of each CVA-classified species to the total value landed in a community by its respective CVA scores for a sensitivity attribute and then summing the resulting values by year, as seen in the following equation:

$$\text{CEVRI}\,(Y_1, \ldots, Y_n) = \sum_{n=1}^{N} a \times b$$

Where,
a = species CVA score
b = % species contribution to total value landed by community

Similarly, CVA-classified species percent contribution to landings was also multiplied by their Total Sensitivity and Total Vulnerability scores to create the Community Total Sensitivity and Community Total Vulnerability indicators. The same

**Table 1. Sensitivity attributes used to calculate three of the Community Environmental Variability Risk Indicators and their respective goals and characterization of low and high scores (Modified from [77]).**

| Sensitivity Attribute | Goal | Low Score | High Score |
|---|---|---|---|
| Stock Size/Status | To determine if the stock's resilience is compromised due to low abundance | Low abundance | High abundance |
| Ocean Acidification | Determine the stock's relationship to "sensitive taxa" | Is not sensitive taxa/ does not rely on sensitive taxa for food or shelter | Stock belong to sensitive taxa |
| Temperature | Known temperature of occurrence or distribution as a proxy for sensitivity to temperature | Species found in wide temperature range/ has a distribution across wide latitudinal range and depths | Species found in limited temperature range/ has a limited distribution across latitude and depths |

"Sensitive taxa," e.g., hard corals, mollusks, calcified algae, and echinoderms, have shown negative effects from ocean acidification.

methods described above were then used to calculate regional scores for comparison purposes, using species contribution to total value landed in each region and sub-region.

All species sensitivity and vulnerability CVA scores are measured on a categorical scale ranging from one to four: (1) low, (2) moderate, (3) high, and (4) very high. The resulting community scores also range between one and four and reflect the relationship between species contributions to value landed in a coastal fishing community and those species' vulnerability to environmental variability. Thus, high percent contributions of highly vulnerable species will result in higher CEVRI scores (e.g., 100% contribution of a species with a very high score (4) for a given sensitivity attribute will result in a community score of very high (4) for that sensitivity attribute: 1 x 4 = 4). Different combinations of percent contribution to landings and species vulnerability scores can result in similar community scores. However, the resulting scores provide a meaningful representation of a community's overall risk to environmental variability factors based on their landings value composition and species dependence and the respective species' bioenvironmental vulnerability (Fig 2).

Although the list of species with CVA scores in the Northeast and Southeast Regions represent a significant portion of the commercial landings in these regions, not all species with regional and local importance were included in the biological assessments, limiting our ability to account for the complete species portfolio for every community. In addition, landings composition varies considerably among different communities in a given region. To ensure that final community scores range between one and four, percent contribution to landings value was calculated based only on the total value of CVA-classified species landed in a community, excluding non-classified species. The five-year average (2018–2022) total percent contribution of classified species by community can be found in the Supplemental Material II.

**Community Level Indicators – Landings Composition Diversity.** Diversity of landings composition is an important factor relating to a coastal fishing community's vulnerability and adaptive capacity, with higher diversity fishery species portfolios being broadly associated with higher adaptive capacity [84]. For the purposes of our analyses, the Simpson's Reciprocal Diversity Index was used, and calculated as 1/D, where:

$$D\left(Y_1, \ldots, Y_n\right) = \sum_{n=1}^{N} \left(\frac{a}{b}\right)^2$$

Where:
a = value landed for a given species
b = total value landed

The Simpson's Reciprocal Diversity Index starts with one as the lowest possible value and ranges to a value that represents the maximum diversity in the sample, thus a higher index value indicates greater diversity. The index accounts

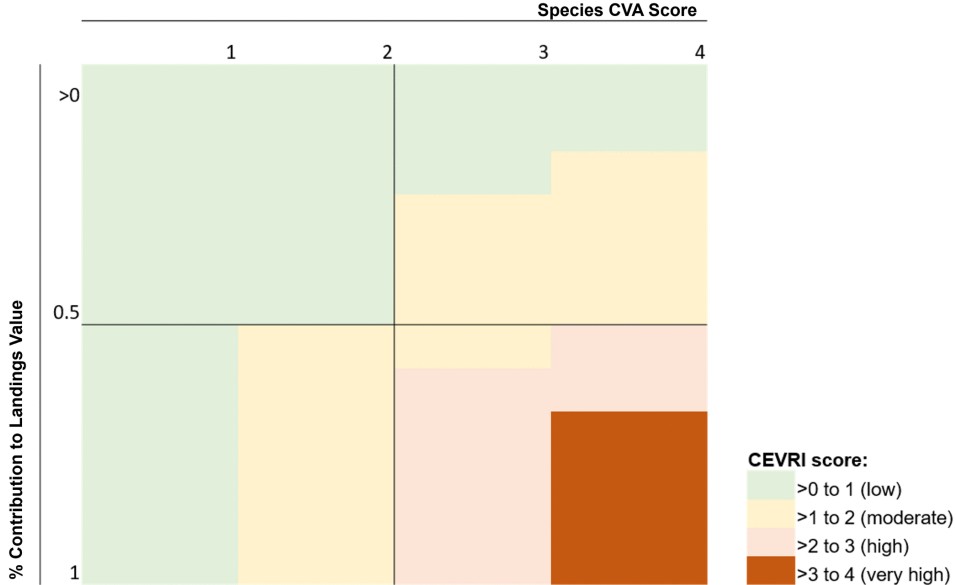

**Fig 2. Visualization of the relationship between species Climate Vulnerability Assessment scores and the percent contribution of a species to value landed in a given community.**

for the relative abundance of each species found in the sample and attributes more weight to more abundant species and vice versa. For this study, the index was calculated for the relative contribution of each species to total value and was used to aid interpretation of results, particularly for longitudinal analysis as part of the coastal fishing community profiles (see Results section).

## Methods of analyses

**Regional level analysis.** Spatial analyses using ArcGIS were conducted to visualize and compare coastal fishing communities in the regions studied with regard to their levels of risk to each of the five fishing community environmental variability indicators: Temperature, Ocean Acidification, Stock Size/Status, Community Total Sensitivity and Community Total Vulnerability. For these analyses, five-year (2018–2022) averages were calculated for each indicator for each coastal fishing community in the regions studied to reflect recent landings composition and minimize the potential effect of annual variation in species landed value.

**Coastal fishing community level analysis.** Risk scores at the fishing community level were analyzed using five-year (2018–2022) averages for each CEVRI along with averages for percent contribution of CVA-classified species for pounds and value, regional quotients for landed pounds and value, and Simpson's Reciprocal Diversity Index scores for pounds and value. Tables showing all communities for each region studied can be seen in Supplemental Material II.

Select communities were also analyzed for longitudinal trends to exemplify the application of the CEVRI for further community-level analyses and profiling. For these analyses, communities with relatively high dependence on classified species for value landed (80% or more) were selected and a complete time series (2000–2022) for each CEVRI was plotted along with regional level scores for comparison purposes. Analysis of landings composition showing percent contribution of each species to total value landed for the same period was used in the interpretation of CEVRI longitudinal trends. The landings composition graphs also include trends in Simpson's Reciprocal Diversity Index calculated as a function of value landed.

## Results

### Regional spatial trends

Figs 3–7 map all coastal fishing communities with landings between 2018 and 2022 color coded for their five-year average score in each of the CEVRI. In general, communities in the Gulf of America and Florida Keys present lower levels of risk to the factors analyzed when compared to the South Atlantic and the Northeast (Figs 3 and 4). Similarly, analyses of landings trends show that the most important species in terms of value landed for the Gulf of America and Florida Keys are species with lower vulnerability to the factors and attributes used in this study when compared to other regions (Supplemental Materials III).

Out of the three separate species sensitivity factors explored in this study, ocean acidification was the one that showed the highest risk for communities. The overall concentration of communities with scores in the high and very high levels is more prominent in the Northeast region and the South Atlantic sub-region. Clusters of very high risk communities are observed particularly in New England (specifically the states of Rhode Island, Massachusetts, New Hampshire, and Maine), the state of Virginia, and Southeast Florida. Top species landed in New England that are relatively vulnerable to ocean acidification include sea scallop, American lobster, eastern oyster (*Crassostrea virginica*), northern quahog

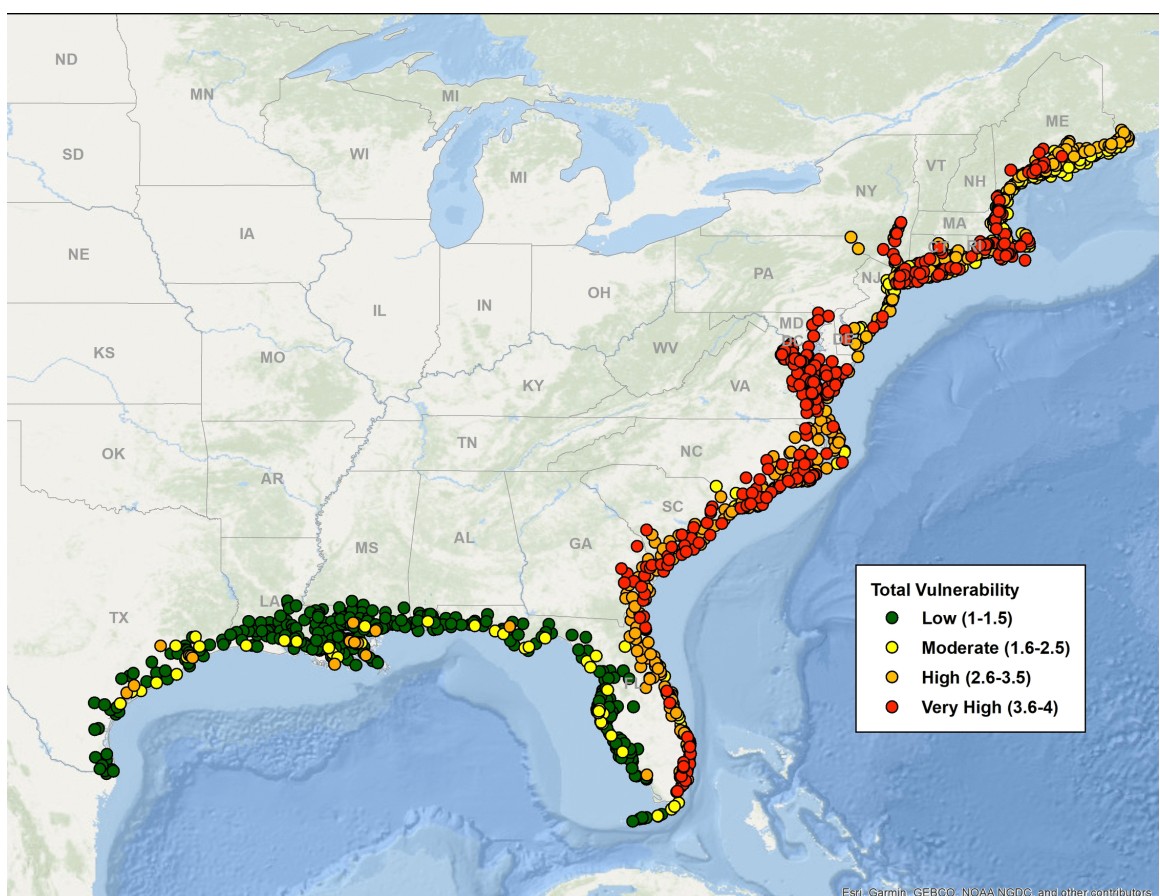

**Fig 3. Map color-coded for Community Total Vulnerability levels from low to very high of all Northeast and Southeast communities with dealer reported landings for species included in the Climate Vulnerability Assessment.** Data source: [21]. The basemap is provided by Esri, GEBCO, NOAA, National Geographic, and other contributors.

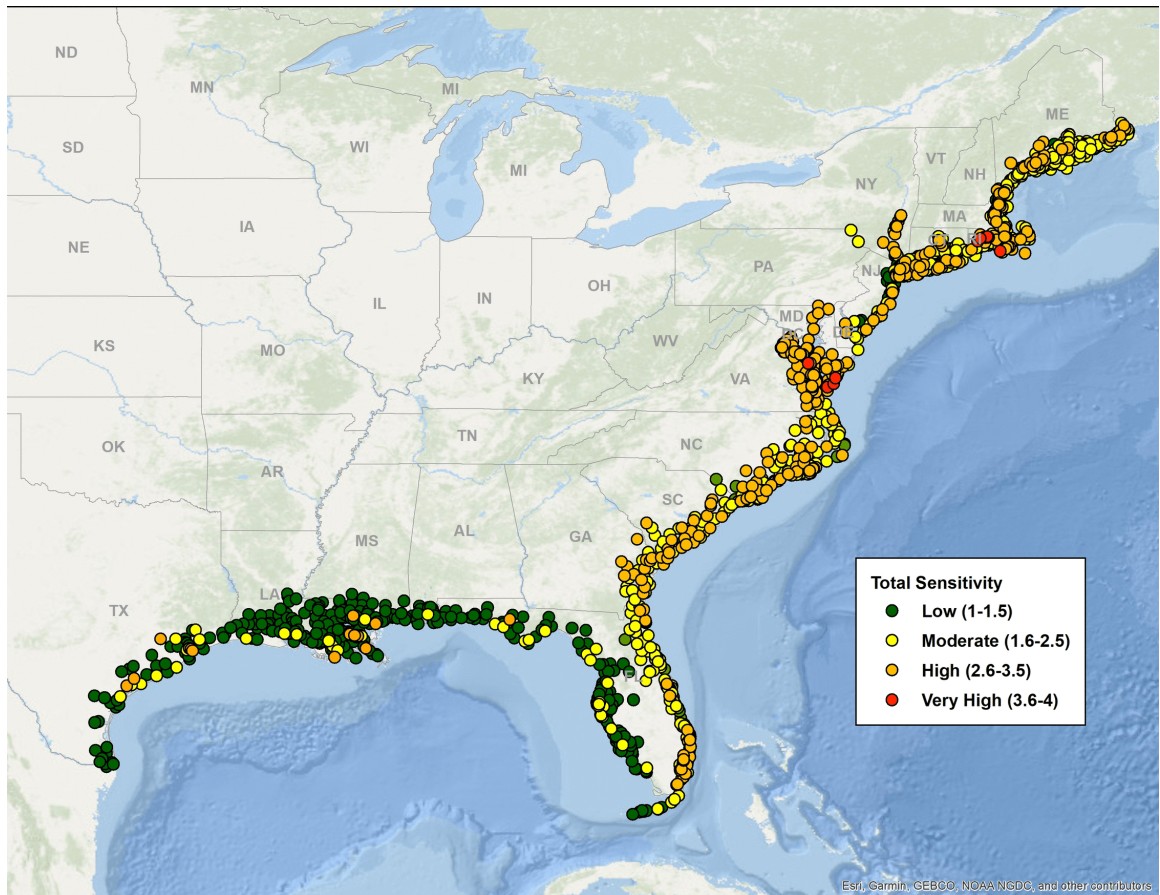

**Fig 4. Map color-coded for Community Total Sensitivity levels from low to very high of all Northeast and Southeast communities with dealer reported landings for species included in the Climate Vulnerability Assessment.** Data source: [21]. The basemap is provided by Esri, GEBCO, NOAA, National Geographic, and other contributors.

(*Mercenaria mercenaria*), haddock (*Melanogrammus aeglefinus*), and summer flounder (*Paralichthys dentatus*). In Virginia, top species include eastern oysters, sea scallop, Atlantic menhaden (*Brevoortia tyrannus*), northern quahog and blue crab (*Callinectes sapidus*). In Southern Florida, communities that appear as highly vulnerable to ocean acidification are primarily dependent upon landings of spiny lobster (*Panulirus argus*). In the Gulf of America, the communities presenting the highest risk to the effects of ocean acidification are located in the states of Louisiana and Texas (Fig 5), where landings of species like eastern oyster, white shrimp (*Litopenaeus setiferus*), brown shrimp (*Farfantepenaeus aztecus*), and blue crab contribute significantly to value landed and present relatively high levels of sensitivity to ocean acidification.

Community risk to species sensitivity to stock size and status also showed some regional variation with a higher number of communities showing high levels of risk in the Northeast region and the South Atlantic sub-region when compared to the Gulf of America and Florida Keys (Fig 6). CEVRI scores for Temperature show relatively uniform distribution of low and moderate levels for all regions studied (Fig 7).

### Regional risk distribution and diversity

Pearson's correlation analysis comparing Community Total Vulnerability and Simpson's Reciprocal Diversity Index scores show that the majority of communities in all regions studied have relatively low diversity of landings composition in terms

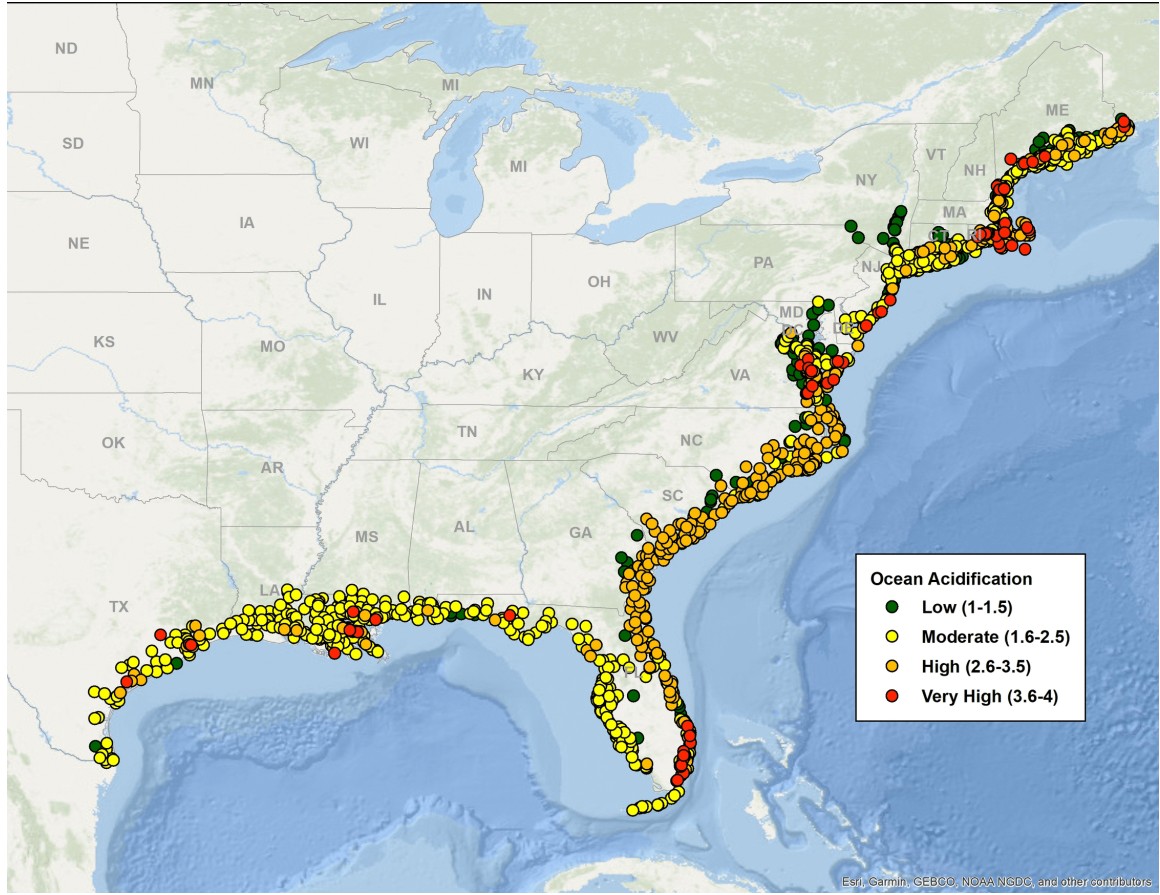

**Fig 5. Map color-coded for the Ocean Acidification indicator five-year average (2018-2022) levels from low to very high of all Northeast and Southeast communities with dealer reported landings for species included in the Climate Vulnerability Assessment.** Data source: [21]. The basemap is provided by Esri, GEBCO, NOAA, National Geographic, and other contributors.

of value, meaning that they are typically reliant on a select few species for their revenue. In the Northeast (Fig 8) and South Atlantic (Fig 9), the majority of communities have above moderate scores for the Community Total Vulnerability indicator, while in the Gulf of America/Florida Keys (Fig 10) the majority of communities have low Community Total Vulnerability scores. Regional differences were also found for the relationship between Community Total Vulnerability and Simpson's Reciprocal Diversity Index scores. In the Northeast and South Atlantic the observed relationship is negative, meaning communities with low landings diversity tend to present higher Community Total Vulnerability scores. In the Gulf of America/Florida Keys, that relationship is positive, with less diverse communities also presenting lower scores for Community Total Vulnerability (Figs 8–10).

## Coastal fishing community profiles

Four communities representing the different regions and sub-regions studied were selected for profiling. Communities selected displayed a high percentage of landings revenue coming from CVA-classified species and a high value contribution to the regional quotient across the 2018–2022 five-year average (Supplemental Material II). These profiles exemplify the use of the CEVRI to analyze trends in coastal fishing communities' level of risk to the environmental variability factors studied and the relationship between risk levels and landings composition.

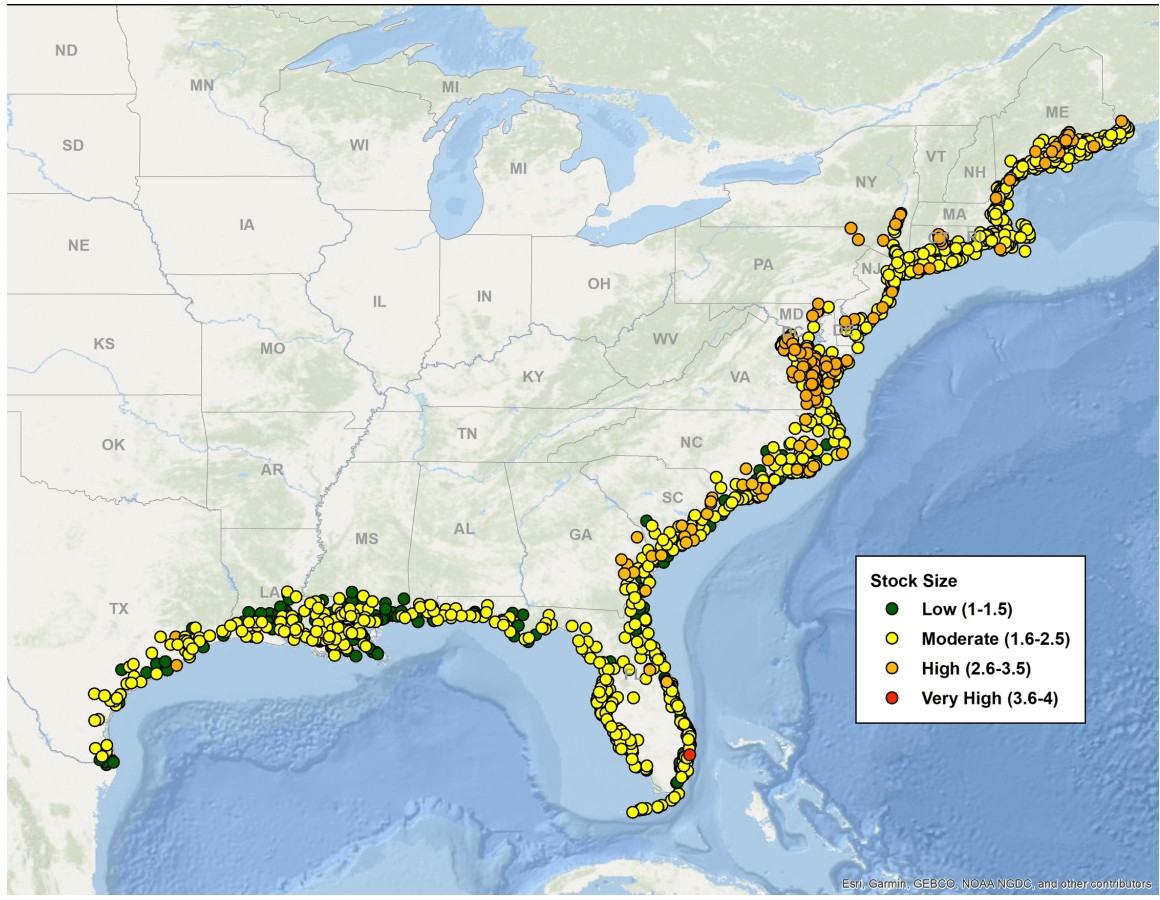

**Fig 6. Map color-coded for the Stock Size/Status indicator five-year average (2018-2022) levels from low to very high of all Northeast and Southeast communities with dealer reported landings for species included in the Climate Vulnerability Assessment.** Data source: [21]. The basemap is provided by Esri, GEBCO, NOAA, National Geographic, and other contributors.

**New Bedford, MA (Northeast Region).** New Bedford, Massachusetts contributed 21% of the 5 year average (2018–2022) regional value quotient of the Northeast, by far the highest in the region, with 98% of the port's landings revenue coming from CVA-classified species on average. As one of the most prominent fishing ports in the United States, New Bedford is renowned for its long-standing commercial fishing sector. In 2022, the port retained its position as the highest-grossing fishing port in the nation, with commercial landings valued at approximately $443 million dollars [85].

New Bedford ranks significantly higher than the Northeast regional average in terms of Ocean Acidification and Temperature, though it scores lower than the regional average for the Stock Size/Status indicator (Fig 11). Overall, Community Total Sensitivity and Community Total Vulnerability scores are both trending above the regional average (Fig 12). As shown in Fig 13, landings in New Bedford are increasingly dominated by the lucrative sea scallop fishery, which now accounts for approximately 85% of total landings revenue. This growing reliance on sea scallops has contributed to a decline in the community's fishery diversity score, which has dropped from over 3.0 in 2000 to under 1.5 in recent years. New Bedford's dependence on sea scallops is reflected in the observed elevated Ocean Acidification indicator scores and moderate scores for the Temperature indicator.

**Narragansett, RI (Northeast Region).** Narragansett, Rhode Island contributed 3% of the 5 year average (2018–2022) regional value quotient for the Northeast, with 96% of the port's landings revenue coming from CVA-classified species

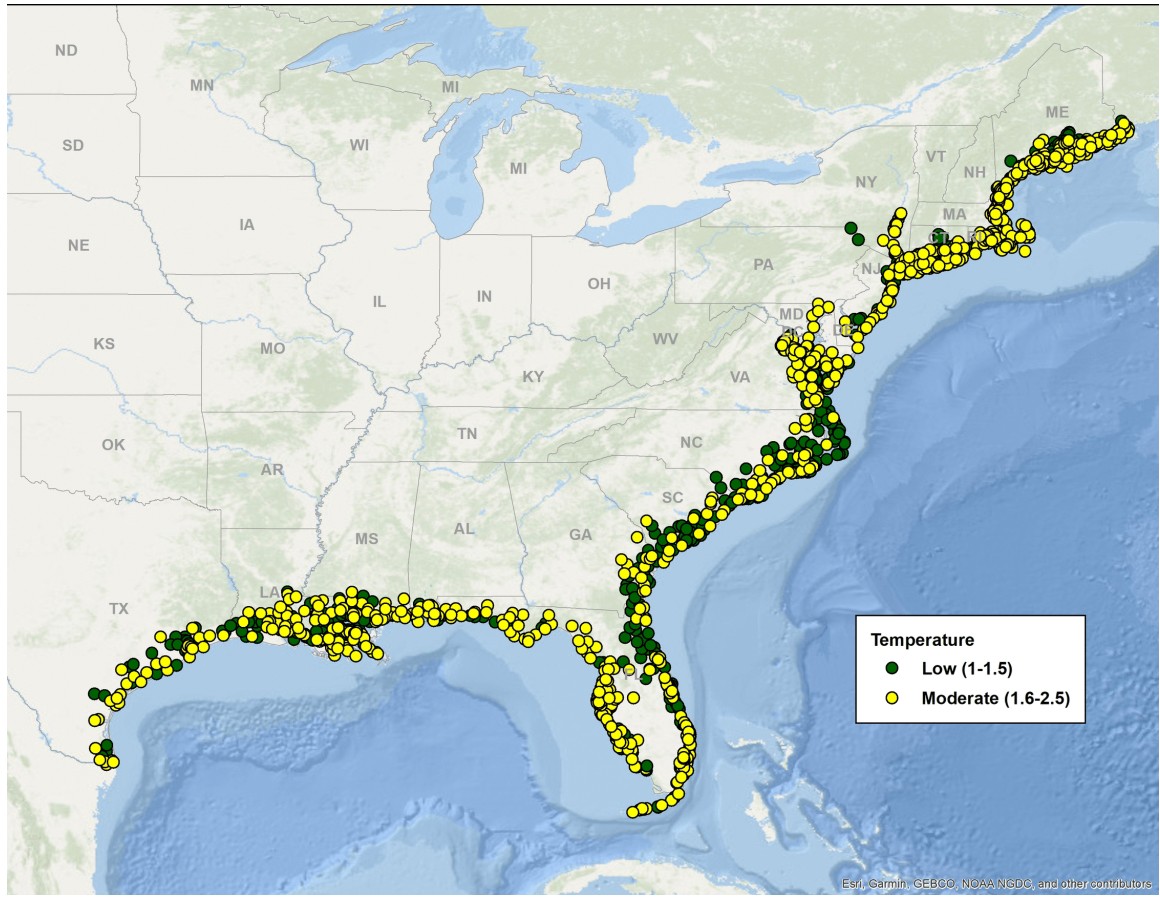

**Fig 7. Map color-coded for the Temperature indicator five-year average (2018-2022) levels from low to very high of all Northeast and Southeast communities with dealer reported landings for species included in the Climate Vulnerability Assessment.** Data source: [21]. The basemap is provided by Esri, GEBCO, NOAA, National Geographic, and other contributors.

on average. The port of Point Judith (located in the town of Narragansett), in particular, has long been a significant hub for the fishing industry and continues to be one of New England's most important fishing centers. In 2022, total landings in Narragansett were valued at approximately $71.4 million, with the seafood processing sector also contributing substantially to the local economy [85].

Overall, Narragansett's CEVRI scores for Ocean Acidification, Temperature, and Stock Size/Status have remained low to moderate and around or below the regional average (Fig 14). Community Total Sensitivity and Community Total Vulnerability scores have also remained relatively low, though they show a slight increase in recent years (Fig 15). As shown in Fig 16, the 2000–2022 period was marked by a significant decline in the lobster fishery, while the prominence of Longfin squid (*Doryteuthis (Amerigo) pealeii*) and sea scallops grew, now accounting for roughly 70% of total landings revenue. Narragansett has typically demonstrated high fishery diversity, reaching a peak score of over 8.0 between 2007 and 2014. Despite declines in recent scores, overall diversity remains relatively high and likely continues to play a key role in keeping Narragansett's CEVRI scores below the regional average, as they have consistently been throughout the time series.

**Atlantic Beach, FL (SE Region – South Atlantic).** Atlantic Beach, Florida contributed 4% of the 5 year average (2018–2022) regional value quotient of the South Atlantic sub-region, with 98% of landings revenue coming from CVA-classified species on average. The community of Atlantic Beach is located just east of

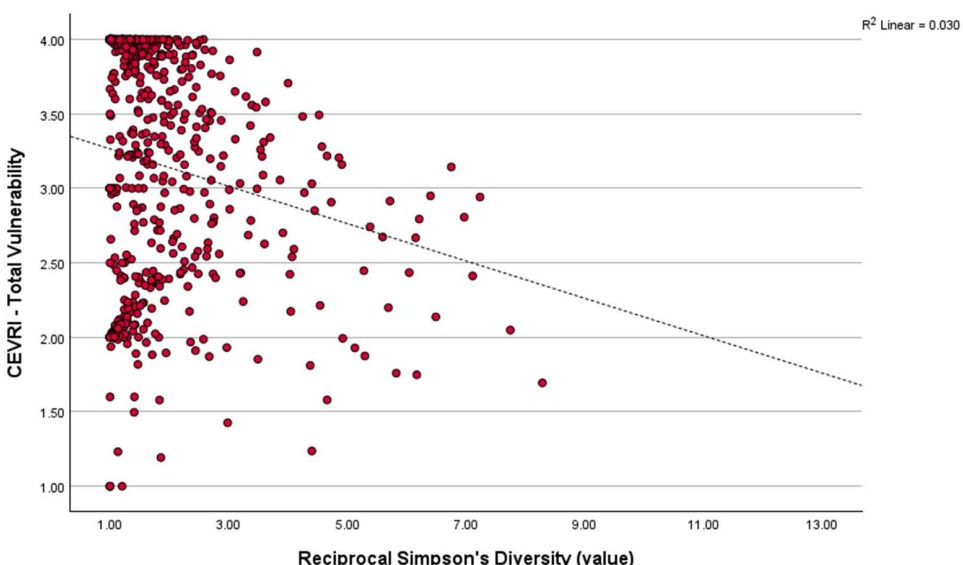

**Fig 8. Scatterplot correlating Community Total Vulnerability indicator and Simpson's Reciprocal Diversity Index five-year average (2018−2022) scores showing the distribution of communities in the Northeast region (P=−0.192, p<0.001).**

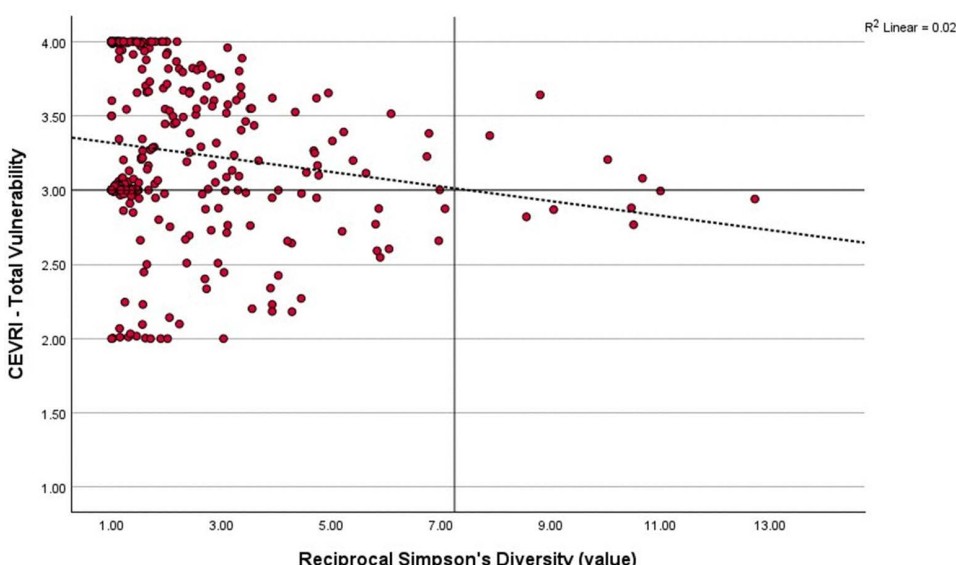

**Fig 9. Scatterplot correlating Community Total Vulnerability indicator and Simpson's Reciprocal Diversity Index five-year average (2018−2022) scores showing the distribution of communities in the South Atlantic sub-region of the Southeast (P=−0.171, p<0.01).**

Jacksonville, FL and includes landings from the historic fishing community of Mayport, situated at the mouth of the St. Johns River. In the last five years, the city of Jacksonville has received funds to restore portions of the working waterfront to preserve the area's fishing heritage [86]. In 2022, commercial landings in Atlantic Beach were valued at approximately $10.4 million, making it one of the top five ports in Florida in terms of landings revenue [85].

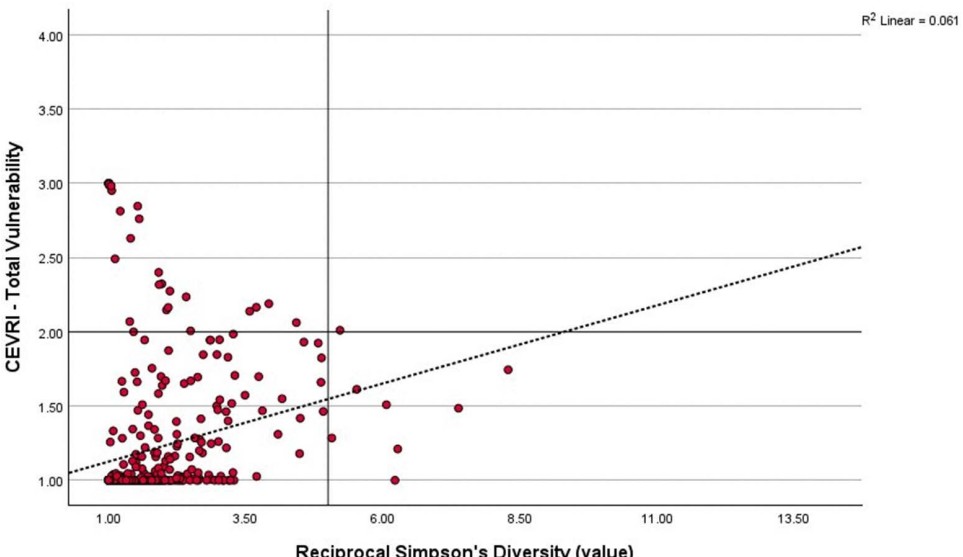

**Fig 10. Scatterplot correlating Community Total Vulnerability indicator and Simpson's Reciprocal Diversity Index five-year average (2018-2022) scores showing the distribution of communities in the Gulf of America/Florida Keys sub-region of the Southeast (P = 0.246, p < 0.001).**

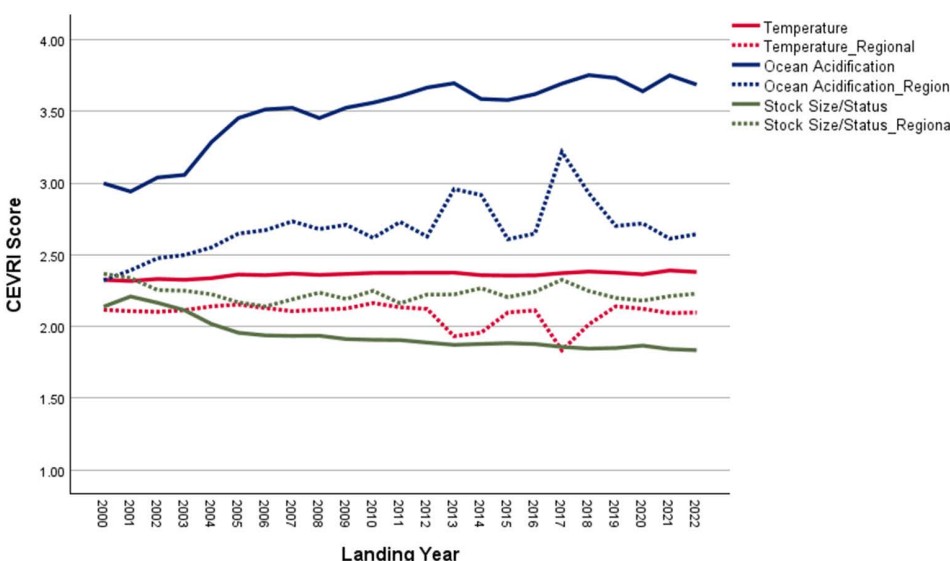

**Fig 11. Yearly scores between 2000 and 2022 for the Ocean Acidification, Temperature, and Stock Size/Status indicators for New Bedford, MA with regional averages for the Northeast.**

Atlantic Beach's scores for the Ocean Acidification and Temperature indicators are high relative to the regional average (Fig 17). Community Total Sensitivity and Community Total Vulnerability scores have also been consistently higher than the South Atlantic regional average throughout the reported time series (Fig 18).

Fig 19 illustrates that the diversity of landings for Atlantic Beach in 2022 was just over 2, dominated by white shrimp. A few other species, including vermilion snapper (*Rhomboplites aurorubens*) and other shellfish are also landed

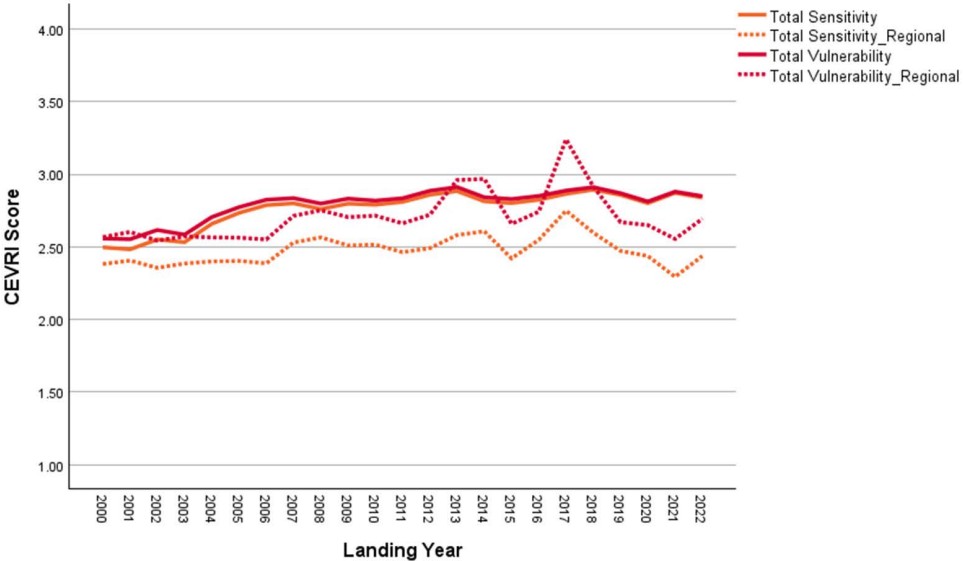

**Fig 12. Yearly scores between 2000 and 2022 for the Community Total Sensitivity and Community Total Vulnerability indicators for New Bedford, MA with regional averages for the Northeast.**

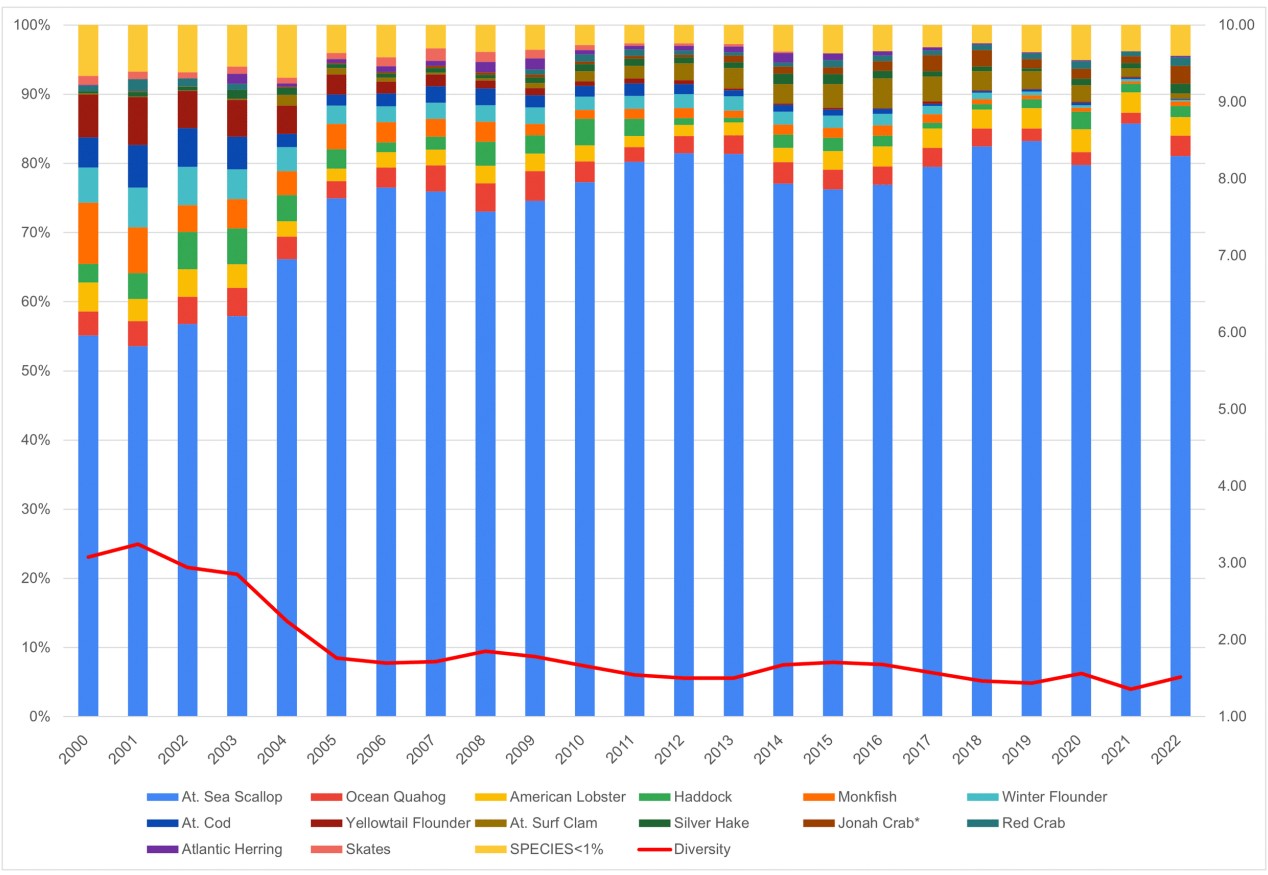

**Fig 13. Yearly landings composition and Simpson's Reciprocal Diversity Index scores based on value between 2000 and 2022 for New Bedford, MA.** *Species not included in the Climate Vulnerability Assessment.

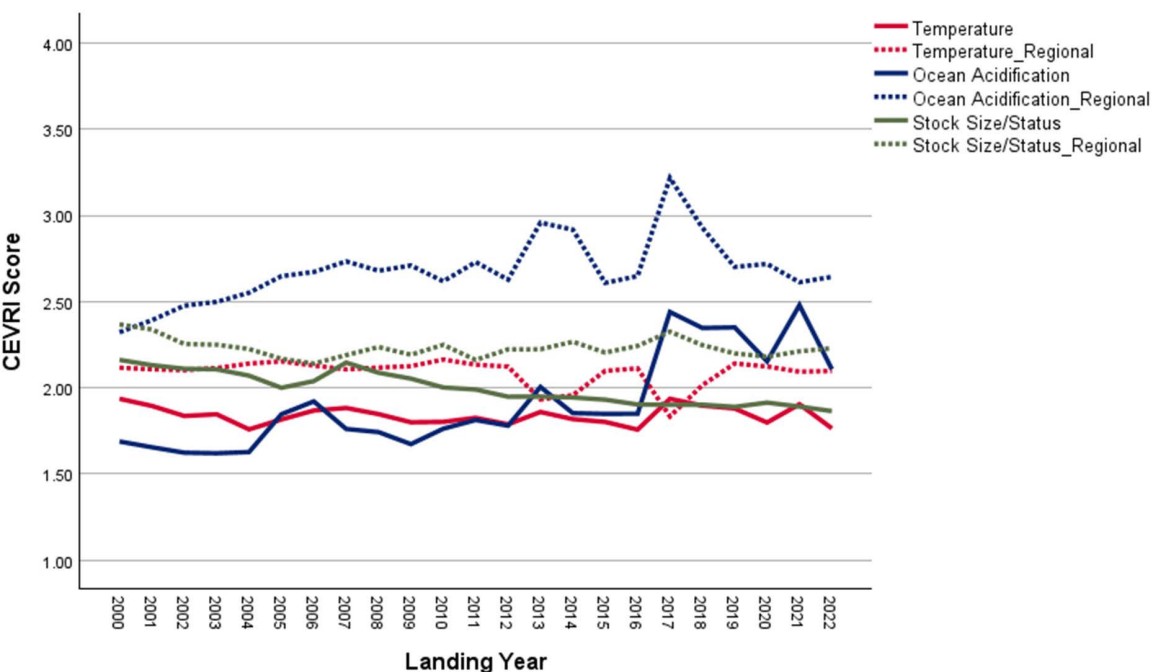

**Fig 14. Yearly scores between 2000 and 2022 for the Ocean Acidification, Temperature, and Stock Size/Status indicators for Narragansett, RI with regional averages for the Northeast.**

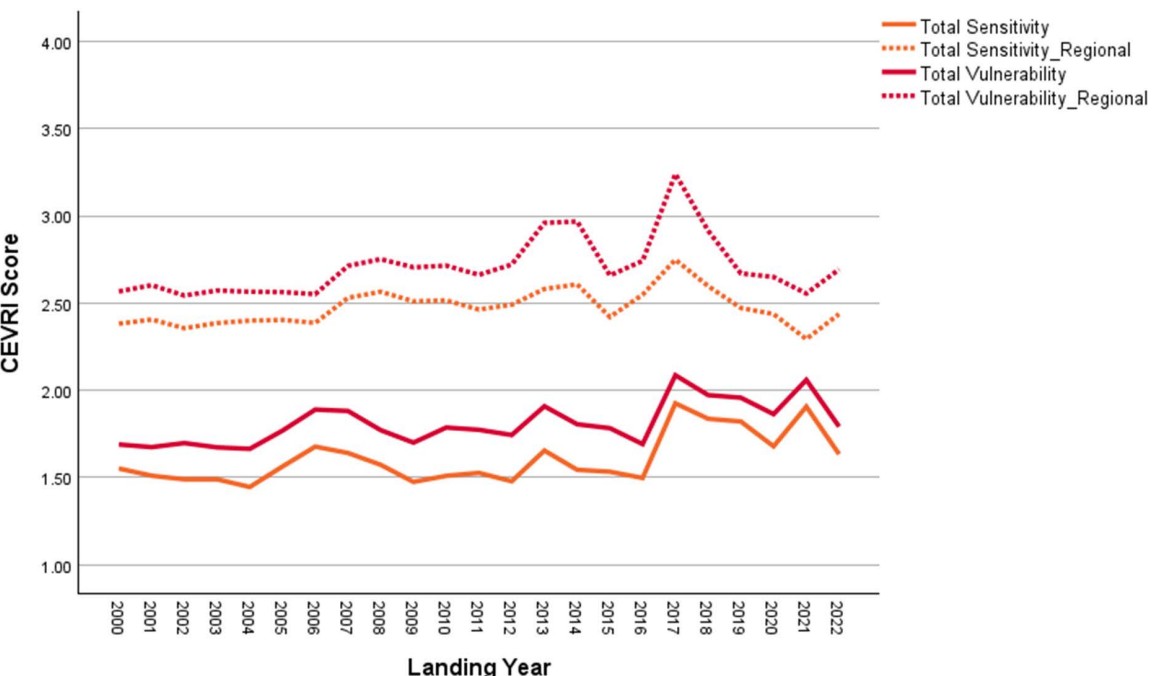

**Fig 15. Yearly scores between 2000 and 2022 for the Community Total Sensitivity and Community Total Vulnerability indicators for Narragansett, RI with regional averages for the Northeast.**

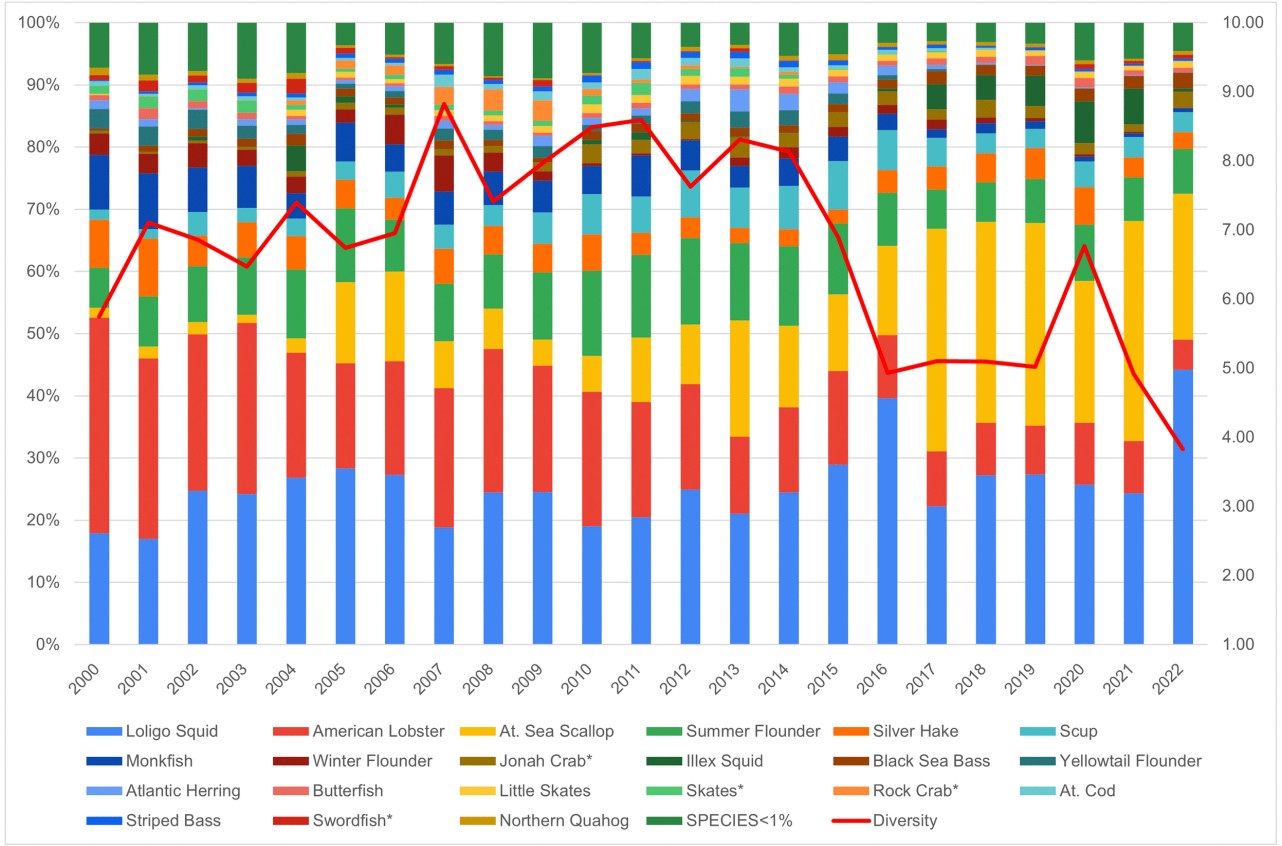

**Fig 16. Yearly landings composition and Simpson's Reciprocal Diversity Index scores based on value between 2000 and 2022 for Narragansett, RI.** *Species not included in the Climate Vulnerability Assessment.

commercially, but these represent a small fraction of overall landings. The diversity of species landed, in terms of value, has fluctuated somewhat year-by-year, largely based on the relative contribution of different shrimp species to overall landings in a given year. However, the overall trend has remained relatively flat throughout the 2000–2022 period, reflecting the community's high dependence on white shrimp. This dependency likely drives the relatively high scores for the Ocean Acidification and Temperature indicators observed for the community.

**Bayou La Batre, AL (SE Region – Gulf of America).** Bayou La Batre, Alabama contributed 5% of the 5 year average (2018–2022) regional value quotient of the Gulf of America sub-region, with 98% of landings revenue coming from CVA-classified species on average. Located in southern Mobile County, just west of Mobile Bay on the Mississippi Sound, Bayou La Batre is a small fishing community with a long history of commercial fishing and seafood processing. In addition to being an important fishing port, the community is home to several large processors that handle significant amounts of shrimp trucked in from other states.

In 2022, commercial landings in Bayou La Batre were valued at approximately $59.5 million, making it one of the top commercial fishing ports in the Gulf [85]. As shown in Fig 20, scores for the Temperature indicator are higher for Bayou La Batre than for other Gulf sub-region communities due to the mix of species landed. Conversely, scores for Ocean Acidification and Stock Size/Status indicators are both below the Gulf of America average. Overall, the Community Total Sensitivity and Community Total Vulnerability scores remain well below the Gulf sub-region average, with little variation over the period between 2000 and 2022 (Fig 21).

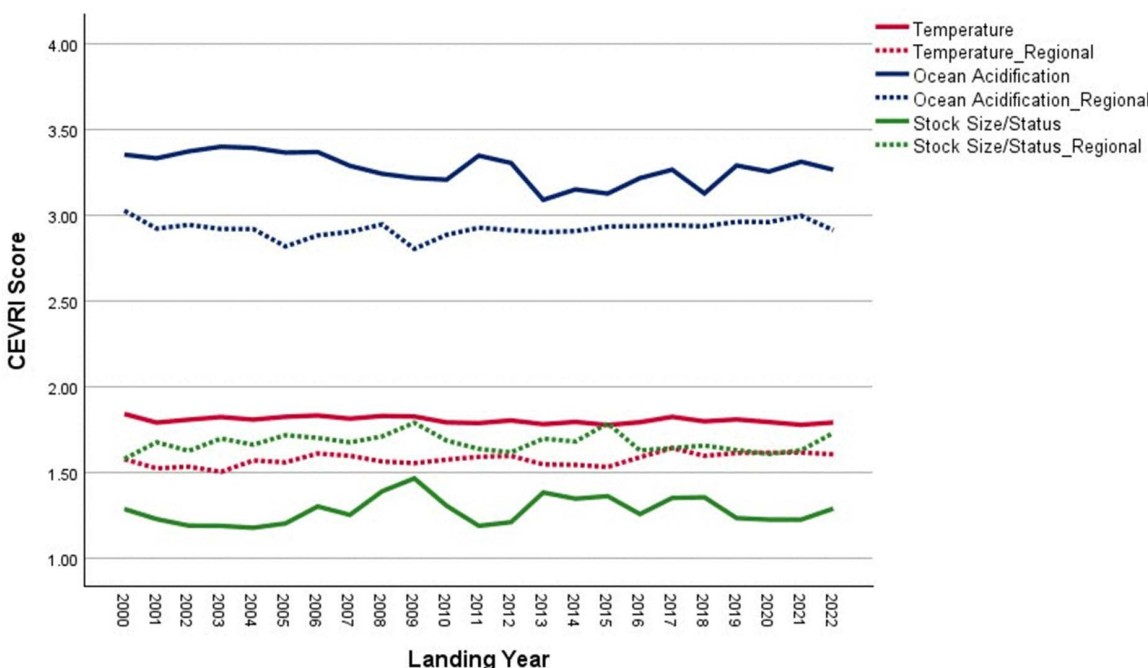

**Fig 17. Yearly scores between 2000 and 2022 for the Ocean Acidification, Temperature, and Stock Size/Status indicators for Atlantic Beach, FL with regional averages for the South Atlantic sub-region of the Southeast.**

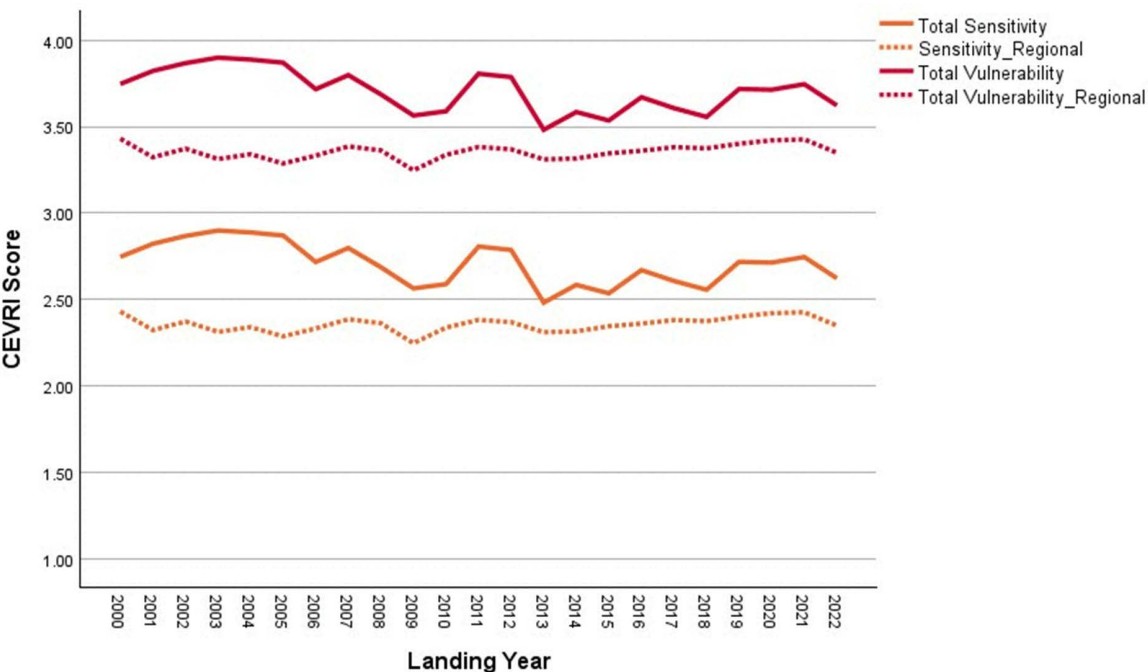

**Fig 18. Yearly scores between 2000 and 2022 for the Community Total Sensitivity and Community Total Vulnerability indicators for Atlantic Beach, FL with regional averages for the South Atlantic sub-region of the Southeast.**

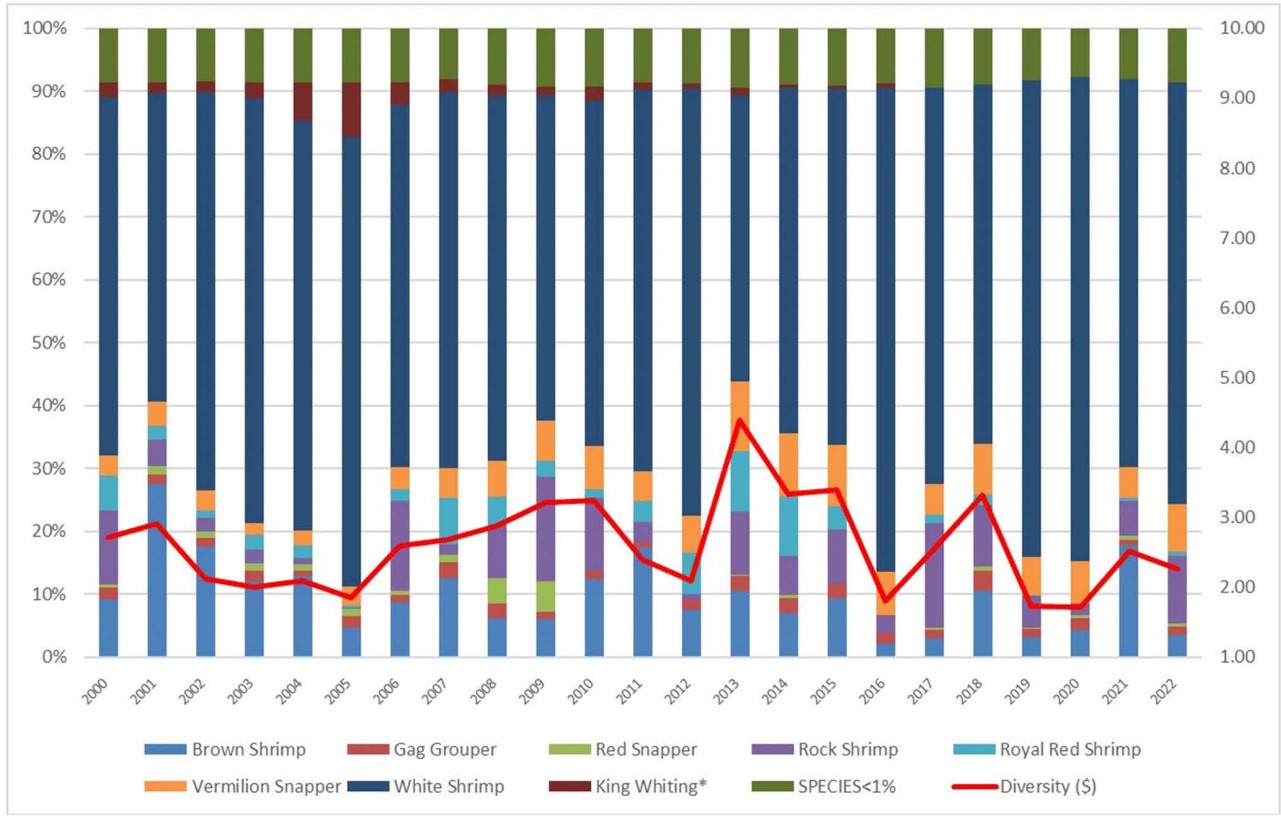

**Fig 19. Yearly landings composition and Simpson's Reciprocal Diversity Index scores based on value between 2000 and 2022 for Atlantic Beach, FL.** *Species not included in the Climate Vulnerability Assessment.

Throughout the 2000–2022 period, Bayou La Batre demonstrated a consistently low diversity of landings, with scores ranging between 2 and 3, primarily due to the dominance of brown shrimp in the landings value (Fig 22). However, in recent years, the diversity score has increased slightly, reflecting the growing importance of pink shrimp (*Farfantepenaeus duorarum*, Penaeidae) and white shrimp relative to brown shrimp. The community's high dependence on shrimp species contributes to its relatively high scores for Temperature, creating potential vulnerabilities under certain environmental variability scenarios.

## Discussion

This study used an interdisciplinary approach to develop indicators of risk to environmental variability at the coastal fishing community level for the Northeast and Southeast regions of the U.S. These indicators were used to classify communities in terms of their risk to three species sensitivity factors: Temperature, Ocean Acidification, and Stock Size/Status, as well as Total Sensitivity and Total Vulnerability. We calculated scores to understand risk as a function of community dependence on species that had been classified in terms of their biological vulnerability to fluctuations in different environmental factors using a system first developed by [55]. The community-level scores were used to spatially analyze and understand risk variability within and between regions. The indicators were also used to create a set of exemplar community profiles to understand temporal trends in risk to the different environmental variability factors as each community's revenue dependence on different species changes through time.

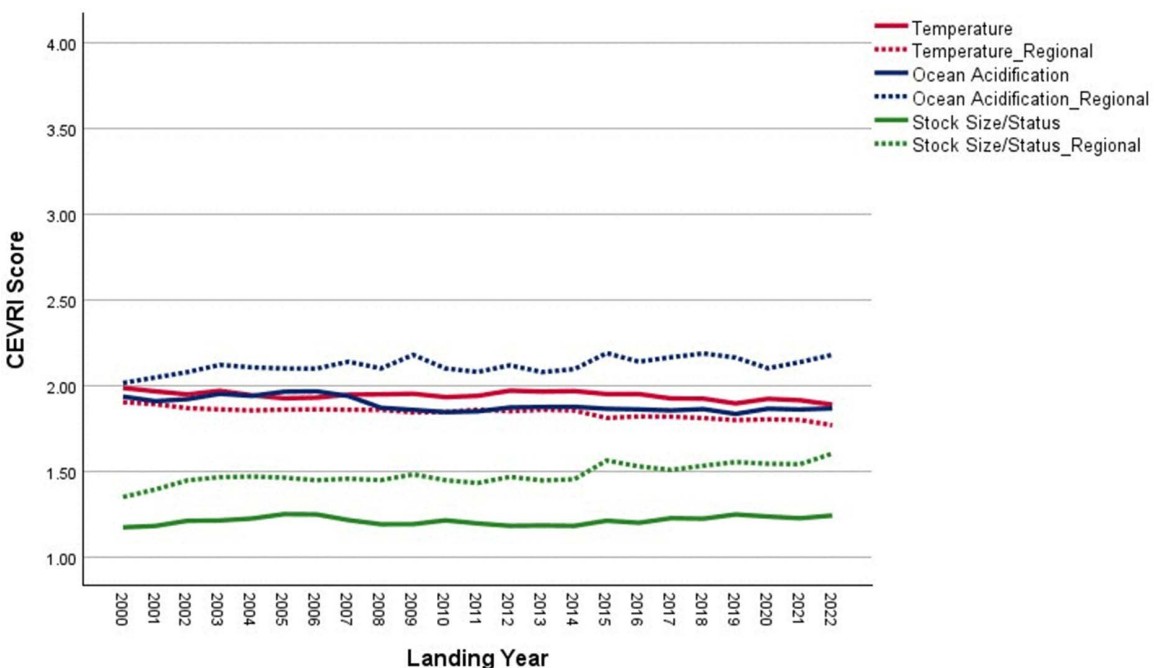

**Fig 20. Yearly scores between 2000 and 2022 for the Ocean Acidification, Temperature, and Stock Size/Status indicators for Bayou La Batre, AL with regional averages for the Gulf of America sub-region of the Southeast (excluding Florida Keys).**

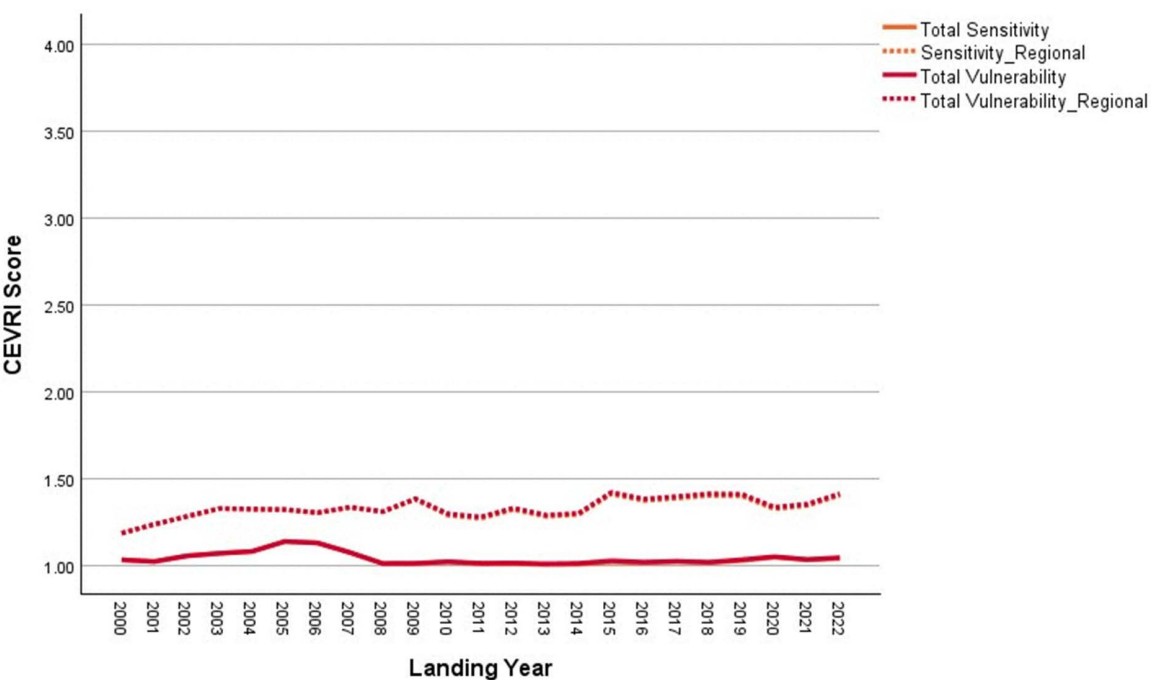

**Fig 21. Yearly scores between 2000 and 2022 for the Community Total Sensitivity and Community Total Vulnerability indicators for Bayou La Batre, AL with regional averages for the Gulf of America sub-region of the Southeast (excluding Florida Keys).**

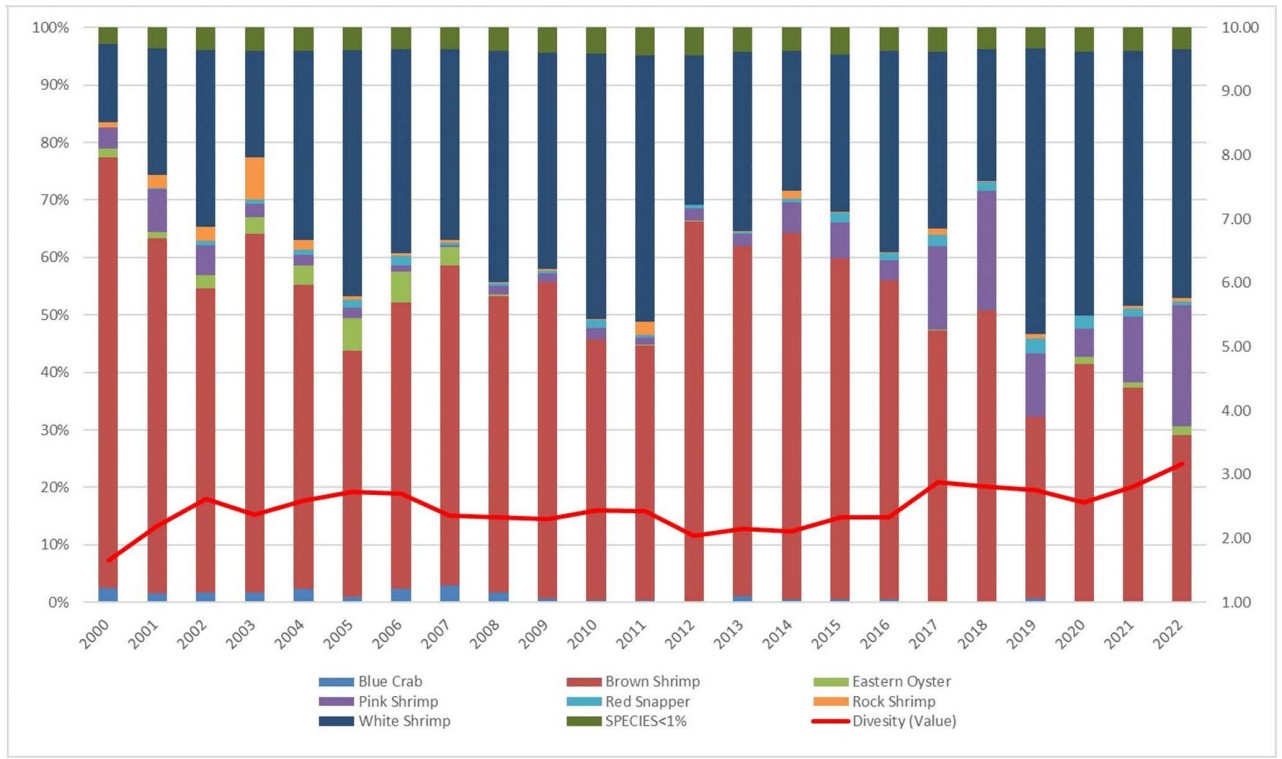

**Fig 22. Yearly landings composition and Simpson's Reciprocal Diversity Index scores based on value between 2000 and 2022 for Bayou La Batre, AL.** *Species not included in the Climate Vulnerability Assessment (excluding Florida Keys).

Spatial analysis of the different indicators developed in this study shows differences between regions with regard to their level of risk to different factors, with the highest disparities observed between both the Northeast and South Atlantic regions relative to the Gulf of America/Florida Keys for all indicators except Temperature. Interestingly, some of the same species landed in the Gulf of America/Florida Keys and the South Atlantic have drastically different CVA scores (e.g., brown shrimp and spiny lobster), emphasizing the importance of regional classification of species vulnerability that takes into consideration region-specific sensitivity and exposure factors affecting different populations. These regional idiosyncrasies are reflected in differences observed in community level risk between Atlantic Beach, FL and Bayou La Batre, AL, both highly dependent upon similar species of shrimp but with different levels of overall risk to the environmental variability factors analyzed.

Analyses at the community level also show that risk to environmental variability factors can increase significantly for communities substantially dependent on fewer highly vulnerable species such as New Bedford, MA with its increasing reliance on sea scallops, a species highly vulnerable to the effects of ocean acidification and moderately vulnerable to increases in temperature. Social vulnerability associated with the potential impacts of ocean acidification for communities highly dependent on shellfish has been extensively discussed by [87]. In contrast, Narragansett, RI, with a more diversified portfolio of landings, has maintained relatively moderate levels of risk through time. However, an increase in the contribution of sea scallops and squid, which led to an overall decrease in revenue diversity, has translated into slightly higher levels of risk to ocean acidification in Narragansett starting in 2016. These results help illustrate and further emphasize the well described relationship between landings composition diversity and overall resilience and adaptive capacity in fisheries [81,88,89]. Our analyses also show that the relationship between risk and diversity of landings composition is

not always straightforward. Communities in the Gulf of America/Florida Keys sub-region with lower diversity scores also tended to present lower environmental variability risk levels for the factors analyzed. Similarly, the Gulf of America profiled community, Bayou La Batre, AL, scored relatively low in terms of environmental variability risk despite being highly dependent on a small number of shrimp species. While the species in question did not show high vulnerability to the factors analyzed, they may be vulnerable to other environmental fluctuations and risks not considered in our analyses, emphasizing the need to couple the CEVRI assessment with other indicators and data streams to provide a more complete picture of community risk to multiple stressors.

The risk associated with fishing community dependence on one or a few high value species has been described by [90] as a "gilded trap." In that study, the authors explain that high market value and economic reliance of Gulf of Maine communities on American lobster has led to a highly specialized fishery consequently more vulnerable to potential stock declines. This risk is well illustrated by the natural resource disaster that occurred in Long Island Sound (LIS) in the late 1990's, devastating the local lobster population and impacting coastal fishing communities [91], with long lasting effects observed even after two decades [92]. While the causes of the LIS die-off have been attributed to a combination of factors that include bottom hypoxia and the presence of pathogens and pollutants, increased water temperature has also been discussed as a significant factor affecting the overall resilience of the LIS lobster population [93–95]. Regardless of the underlying factors leading to the disaster, the LIS lobster die-off is a telling example of the risks faced by coastal fishing communities highly dependent upon one or few species. The distribution patterns observed in this study suggest that the majority of communities in the regions studied present relatively low revenue portfolio diversity. Particularly in the Northeast region and South Atlantic sub-region, the majority of these communities also present high scores of overall risk to environmental variability based on sensitivity and exposure assessments of the species primarily contributing to their total revenue. The relationship between landings composition diversity and risk to impacts such as those associated with the effects of environmental variability is a pertinent topic in the context of management decisions that may affect the ability of fishers to diversify their catch, including species-specific permits and other license restrictions. As fishers become more specialized and may be limited in their choices to exploit alternative species, their risk to the impacts of environmental change become potentially more severe [96]. Thus, consideration of the socio-economic impacts of management decisions can benefit from analyses of associated environmental variability risk factors for individuals and communities. These analyses can also be helpful to communicate risks to broader audiences, particularly with regards to understanding the complex trade-offs facing society when decisions are made to address large-scale impacts such as global environmental changes. For instance, the understanding and visualization of the increasing risk of fishing communities to the effects of ocean acidification, as illustrated by this study's analyses, can be a powerful tool to communicate to coastal fishing communities the need for increasing investment in competing uses of the oceans such as marine development (e.g., offshore energy and marine carbon dioxide capture). Community environmental variability risk indicators can also be cross-referenced with other community-level indicators to identify intersections between environmental fluctuations and other factors, such as social and economic vulnerability [97].

While the CEVRI are limited to assessing risk to specific environmental variations, communities are affected by a number of concurrent stressors that may compound the effects of the disturbances considered, such as pollution and habitat degradation. Future research considering these added stressors and potential interactions between them and those considered in the CVAs would further elucidate risks to community well-being. In addition, research considering factors influencing a community's adaptive capacity in the face of the environmental variations captured by the CEVRI is also a necessary step for further understanding community vulnerability in a more holistic and pragmatic way. Comparing the assessment of potential risk as indicated by the CEVRI to factors that may help a community minimize said risks will provide valuable information about policy and management priorities in the face of specific environmental disturbances. The approach used in this study may also serve as a good model for the development of other indicators to understand the risk of coastal fishing communities to the aforementioned and other competing uses, thus further developing the ability

of multiple stakeholders to understand and assess cumulative impacts and complex trade-offs affecting the sustainability of marine ecosystems and resources.

## Supporting information

**S1 File. Sensitivity and exposure factors used in the Climate Vulnerability Assessments.**
(PDF)

**S2 File. Five-year average values for all communities.**
(PDF)

**S3 File. Regional landings composition 2018–2022.**
(PDF)

## Acknowledgments

We thank Jon Hare, John Quinlan, Todd Kellison, and Michael Burton for providing data and expertise on the species Climate Vulnerability Assessments for their respective regions. We thank Sarah Gaichas for her thoughtful review of the manuscript and ideas for visualization and application of the CEVRI in a management context. Acknowledgment of the above individuals does not imply their endorsement of this work; the authors have sole responsibility for the content of this contribution. The views expressed herein are those of the authors and do not necessarily reflect the views of NOAA or any of its sub-agencies.

## Author contributions

**Conceptualization:** Tarsila Seara, Matthew McPherson, Patricia M Clay, Michael Jepson, Angela Silva.

**Data curation:** Tarsila Seara, Michael Jepson, Changhua Weng.

**Formal analysis:** Tarsila Seara, Changhua Weng.

**Funding acquisition:** Matthew McPherson, Lisa L Colburn.

**Investigation:** Tarsila Seara, Matthew McPherson, Patricia M Clay, Michael Jepson, Lisa L Colburn.

**Methodology:** Tarsila Seara, Michael Jepson.

**Project administration:** Tarsila Seara, Matthew McPherson.

**Resources:** Tarsila Seara, Matthew McPherson.

**Supervision:** Tarsila Seara, Lisa L Colburn.

**Validation:** Tarsila Seara, Patricia M Clay, Michael Jepson, Lisa L Colburn, Angela Silva.

**Visualization:** Tarsila Seara, Changhua Weng.

**Writing – original draft:** Tarsila Seara, Matthew McPherson, Patricia M Clay, Michael Jepson, Lisa L Colburn, Changhua Weng, Angela Silva.

**Writing – review & editing:** Tarsila Seara, Matthew McPherson, Patricia M Clay, Michael Jepson.

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
