## [Decision Letter · Decision Letter 0]

16 Apr 2025

Dear Dr. Seara,

Thank you for submitting your manuscript to PLOS ONE. After careful consideration, we feel that it has merit but does not fully meet PLOS ONE’s publication criteria as it currently stands. Therefore, we invite you to submit a revised version of the manuscript that addresses the points raised during the review process.

We look forward to receiving your revised manuscript.

Kind regards,

Tzen-Yuh Chiang

Academic Editor

PLOS ONE

“Funding for this project was provided by the NOAA Fisheries Office of Science and Technology under the Inflation Reduction Act (IRA).”

“Funding for this project was provided by the NOAA Fisheries Office of Science and Technology under the Inflation Reduction Act (IRA). We thank Jon Hare, John Quinlan, Todd Kellison, and Michael Burton for providing data and expertise on the species Climate Vulnerability Assessments for their respective regions. We thank Sarah Gaichas for her thoughtful review of the manuscript and ideas for visualization and application of the CEVRI in a management context. Acknowledgment of the above individuals does not imply their endorsement of this work; the authors have sole responsibility for the content of this contribution. The views expressed herein are those of the authors and do not necessarily reflect the views of NOAA or any of its sub-agencies. The authors have no conflict of interest to declare.”

“Funding for this project was provided by the NOAA Fisheries Office of Science and Technology under the Inflation Reduction Act (IRA).”

4. We note that Figures 1, 3, 4, 5, 6, and 7 in your submission contain [map/satellite] images which may be copyrighted. All PLOS content is published under the Creative Commons Attribution License (CC BY 4.0), which means that the manuscript, images, and Supporting Information files will be freely available online, and any third party is permitted to access, download, copy, distribute, and use these materials in any way, even commercially, with proper attribution. For these reasons, we cannot publish previously copyrighted maps or satellite images created using proprietary data, such as Google software (Google Maps, Street View, and Earth). For more information, see our copyright guidelines: http://journals.plos.org/plosone/s/licenses-and-copyright.

1. You may seek permission from the original copyright holder of Figures 1, 3, 4, 5, 6, and 7to publish the content specifically under the CC BY 4.0 license. 

Reviewers' comments:

Reviewer's Responses to Questions

**Comments to the Author**

1. Is the manuscript technically sound, and do the data support the conclusions?

Reviewer #1: Yes

Reviewer #2: Partly

2. Has the statistical analysis been performed appropriately and rigorously?

Reviewer #1: I Don't Know

Reviewer #2: No

3. Have the authors made all data underlying the findings in their manuscript fully available?

Reviewer #1: Yes

Reviewer #2: No

4. Is the manuscript presented in an intelligible fashion and written in standard English?

Reviewer #1: Yes

Reviewer #2: Yes

Reviewer #1: This paper includes an impressive amount of work and includes a lot of great information about risk and vulnerability to fishing communities and fisheries species on the U.S. Northeast and Southeast. It is also very well written and easy to follow. The addition of scores overtime is an important, novel contribution. I do however feel there is more work that could be done to strengthen and clarify this manuscript, mainly in the methods.

METHODS

1. Use of the term “vulnerability” and no adaptive capacity indicators – the term Total Vulnerability is used but according to line 199, the Total Vulnerability score is only made up of sensitivity and exposure (“considering all 12 sensitivity and 12 exposure factors”) and doesn’t include adaptive capacity. Though the intergovernmental panel on climate change has changed its definition of vulnerability over time, I’m pretty sure in every iteration it includes some form of adaptive capacity. Based on IPCC and other literature, vulnerability is commonly a combination of sensitivity, exposure, and adaptive capacity. Does the Total Vulnerability indicator used include adaptive capacity? If not, why not and should this then be “risk”?

Furthermore, the introduction on lines 144-149 talks about the importance of considering social factors in combination with environmental factors and also adaptive capacity in general. Yet, I don’t think any social adaptive capacity is looked at within the metric of vulnerability used in this manuscript. Total species diversity is looked as a proxy for adaptive capacity, but what about social components that make up these communities? Yes you found lower risk for the South Atlantic compared to Northeast but what about the variation in social factors between these regions that may make one more able to cope then another? Is it possibly to look at how social adaptive capacity factors would impact your results? And/or at least discuss?

2. Lines 182-184 – By using landed value dollars instead of weight, how much does this change what the top species are? Is the high value for certain species potentially due to high value for a rare species and then the top species don’t actually reflect what is most commonly fished for? In other words, would weight actually reflect what is a more common resource across many fishers? For example, Koehn et al. 2022 looked at – “We focused on the top 90% of landed weight, not revenue, to focus on the majority of what is caught in the community (versus a rare but highly valued species caught).”

3. Lines 219-221 – I know that the Total Sensitivity and Total Vulnerability were taken from another reference, but how were these calculated? How were each of the 12 sensitivity/exposure factors combined and then how was exposure and sensitivity combined? The algorithm for this could change the results.

4. Figure 2 – Flip the y axis so it goes from 0/<0 at the left bottom corner and reads from low to high from up to down. Not sure why the y axis is flipped and goes from high to low

5. Lines 238-240 – Why were certain species of importance not included? How many communities was the full suite of catch not included?

6. Line 272 – The years 2018-2022 were included, did the beginning of covid show any large changes in fishing catch? Was 2020 significantly different than the other years?

RESULTS

7. Figure 6 has one red community – no red on the legend and what community is this? Why so high for that one community?

8. Figures 8-10 – combine into one plot with 3 panels?

9. Lines 332-343 and figures 8-10 – What statistics were used to determine that these trends were significant? Could try quantile regression.

10. I find the community profiles interesting. It does make the paper longer but it’s interesting to draw out these examples. Do most communities fit into similar profiles as the 4 you pulled out?

DISCUSSION

11. Lines 504-505 – You mention how social vulnerability and ocean acidification has been discussed elsewhere. What did that paper find? How would that impact your results given there is probably differences in social vulnerability across the regions you looked at?

12. Lines 518-520 – This is a good point that there may be other environmental factors not considered here that species may be vulnerable to. Have other papers considered these? Also, you note in the methods that Colburn et al. 2016 looked at similar regions – what were their findings compared to yours?

13. Why do you think the one region’s species diversity vs. vulnerability trend was opposite from the other regions and opposite from what’s expected?

Reviewer #2: Review report

This manuscript entitled “Developing environmental variability risk indicators for fishing communities” presents the development and application of Community Environmental Variability Risk Indicators (CEVRI) to evaluate the vulnerability of U.S. Northeast and Southeast fishing communities to environmental variability. Using commercial landings data (2000–2022) and Climate Vulnerability Assessment (CVA) scores for species, the study aims to assess community-level risk based on dependency on vulnerable species. It is an important and timely contribution, particularly in the context of climate adaptation planning for coastal communities. However, the manuscript suffers from critical methodological, conceptual, and analytical limitations that must be addressed before it can be considered for publication. These include insufficient theoretical framing of “risk,” absence of validation or statistical testing, over-simplified treatment of diversity and adaptive capacity, and vague policy relevance. Please, see my detailed comments below.

Major comments

Introduction: The manuscript lacks a formal, operational definition of 'risk'. The distinction between risk, vulnerability, exposure, and adaptive capacity is not clearly articulated. Please, define 'risk' explicitly based on existing frameworks (e.g., IPCC), and clarify how CEVRI maps onto these components.

Lines 182-184: It`s stated that “The species’ landed value in dollars was used, instead of weigh in pounds, to relate risk to environmental variability to community economic dependence on vulnerable species.” The use of commercial landings value (rather than biomass or catch volume) as the sole economic proxy for community dependence on species introduces market volatility as a confounding factor, since price fluctuations may not reflect ecological vulnerability. Therefore, a sensitivity analysis to test how this proxy distorts risk scores under price shifts is required.

Lines 196-201: Table 1 lists three sensitivity attributes used to calculate Community Environmental Variability Risk Indicators (CEVRI), but this requires clarification. While the original Climate Vulnerability Assessment (CVA) framework includes 12 sensitivity and 12 exposure attributes, the current manuscript employs only three sensitivity attributes. This drastic reduction risks oversimplifying species-level risk assessments, particularly in dynamic environments where multiple stressors interact. For instance, excluding exposure factors (e.g., hypoxia, salinity shifts) limits ecological realism by failing to capture critical threats that compound vulnerability. Please explicitly justify this selective use of attributes in the methodology. If driven by data limitations, acknowledge this constraint and discuss how it may affect the interpretation of results (e.g., underestimating risks for species sensitive to omitted stressors). Additionally, consider addressing whether the retained attributes sufficiently represent the full spectrum of climate impacts or inadvertently bias outcomes toward specific vulnerability pathways.

Line 200: I strongly recommend including a comprehensive supplementary table listing all analyzed species with their associated CVA scores, organized by: geographic region, taxonomic group (e.g., hard corals, mollusks, calcified algae), scientific names, and species importance (commercially, recreationally, and ecologically important). This should be accompanied by each species' percentage contribution to total landed value, with special designation for non-CVA assessed species. Crucially, the table should explicitly quantify what percentage of total landings were excluded from analysis per region/community due to lack of CVA scores. Providing these details would serve four key purposes: first, it would enhance methodological transparency by making the study's taxonomic coverage explicit; second, it would enable more robust assessment of regional and taxonomic vulnerability patterns; third, it would facilitate meaningful comparative analyses with other studies; and fourth, it would significantly improve the reproducibility of the findings. The absence of such fundamental data currently limits readers' ability to evaluate potential biases introduced by species exclusions or taxonomic coverage gaps.

Lines 214: In the CEVRI equation, the manuscript should clarify whether species were weighted equally in the calculations or if adjustments were made to account for the disproportionate importance of high-value or dominant species. This distinction is critical for proper interpretation, as equal weighting could mask vulnerability hotspots concentrated in economically significant taxa.

Lines 212-216: The presentation of CEVRI scores as point estimates without confidence intervals or uncertainty measures (Figures 11-12) limits interpretation of result robustness. I recommend incorporating statistical uncertainty quantification through methods like bootstrap or Monte Carlo simulations, particularly since two key inputs introduce variability: (1) expert-derived CVA scores with undocumented weighting criteria, and (2) fluctuating market prices. A sensitivity analysis testing how CEVRI ranks respond to variations in these inputs would greatly strengthen the analysis by identifying which components drive most uncertainty. This is especially important given the policy implications of vulnerability assessments, where understanding score stability across plausible parameter ranges is crucial for decision-making. For a robust bootstrap analysis to quantify uncertainty, I recommend consulting Burton et al. (2023), who demonstrate applicable methods for propagating uncertainty in assessment of overall climate vulnerability (page 7).

Lines 219-222: The methodology for calculating total sensitivity and total vulnerability requires clarification, as an apparent inconsistency exists between the described approach and the implementation. While the text indicates that 12 sensitivity and 12 exposure factors were used (Lines 198-199), the CEVRI values were ultimately derived from only three core sensitivity attributes. To resolve this discrepancy and ensure methodological transparency, three key clarifications are needed. First, the relationship between the 12 factors and the three CEVRI attributes should be explicitly stated, explaining how the broader set of factors maps onto the simplified framework. Second, the mathematical formulation used to aggregate the factors and calculate the total sensitivity and vulnerability metrics should be provided, including all relevant equations or decision rules. Third, the weighting scheme applied to the factors that requires clarification - whether equal weights were used throughout or if differential weighting was employed, along with the rationale for this choice. Providing these details would significantly strengthen the methodological rigor and reproducibility of the analysis.

Line 243: The limited species coverage (71-82 per region) may bias risk assessments by excluding locally important taxa. Please: quantify excluded species, discuss potential impacts on results (e.g., for specialized fisheries), and consider proxy methods for unassessed species (e.g., taxonomic relatives). A supplementary table listing all analyzed species with their CVA scores would improve transparency.

Line 249: The manuscript's use of Simpson’s Reciprocal Index as a proxy for adaptive capacity warrants further discussion given its limitations in capturing key ecological dimensions of resilience. While species diversity metrics provide some insight into portfolio effects, the current approach overlooks two critical factors that mediate true adaptive capacity: functional redundancy (whether landings comprise species with similar ecological roles) and trophic breadth (the range of feeding strategies represented). This is particularly important because a fleet landing diverse but uniformly climate-vulnerable taxa (e.g., multiple mollusk or shrimp species) would appear resilient under the current metric while actually facing compounded risks. I recommend the authors explicitly address this limitation in the discussion, acknowledging how the "diversity equals resilience" assumption may break down when portfolios contain taxonomically clustered, similarly sensitive species. A valuable extension would be developing a weighted diversity index that incorporates species-specific vulnerability scores, where the contribution of each species to the overall index is adjusted by its climate sensitivity. This refinement could better distinguish between truly resilient portfolios (diverse across vulnerability categories) and pseudo-resilient ones (diverse but uniformly vulnerable). At minimum, the discussion should explore how the current metric might overestimate adaptive capacity in regions specializing in taxonomically narrow but species-rich fisheries, with potential examples from the study system.

Line 268-270: The use of five-year aggregated data for indicator time series raises concerns about temporal resolution adequacy in capturing climate vulnerability dynamics. While this approach may smooth interannual variability, it risks obscuring three critical elements: first, the immediate impacts of acute environmental or economic shocks that often drive rapid changes in fishery systems; second, the potentially important short-term adaptive responses communities employ following such disturbances; and third, early warning signals that typically manifest at finer temporal scales before establishing longer-term trends. This resolution limitation is particularly problematic for climate vulnerability assessments where timely detection of emerging risks is paramount for effective adaptation planning. I strongly recommend the authors either analyze the data at annual resolution to preserve these critical signals or provide a robust methodological justification for the five-year aggregation choice - for instance, by demonstrating through sensitivity analyses that key patterns remain detectable at this timescale. A comparison of annual versus aggregated trends in selected high-variability regions would significantly strengthen the manuscript by either validating the current approach or revealing important masked dynamics. Such validation seems especially warranted given that many climate adaptation policies operate on annual decision cycles where missing short-term vulnerability fluctuations could lead to suboptimal management interventions.

Policy relevancy: The manuscript’s policy relevance is significantly undermined by the absence of actionable guidance for translating CEVRI scores into tangible management interventions. While the framework is presented as a tool for climate adaptation, it does not clarify how stakeholders—such as fishery managers, disaster relief agencies, or conservation planners—should operationalize these vulnerability rankings in practice. For instance, there is no discussion of how CEVRI could inform dynamic quota adjustments during climate anomalies, prioritize infrastructure upgrades in high-risk communities, or trigger preemptive disaster relief for regions crossing specific vulnerability thresholds. This gap is compounded by the lack of empirical validation against real-world outcomes (e.g., historical fishery collapses, post-disaster recovery rates), leaving uncertainty about whether CEVRI scores reliably predict risks or merely reflect theoretical constructs. Furthermore, the oversimplified use of three sensitivity attributes (versus the original CVA’s 12) and exclusion of critical exposure factors like hypoxia or salinity shifts risks misrepresenting species-level vulnerabilities, particularly in ecosystems facing multi-stressor interactions. These methodological choices, combined with unaddressed data gaps (e.g., unreported landings, unvalidated expert judgments), raise questions about the framework’s ecological realism and its capacity to inform place-based management. To bridge these gaps, the authors must explicitly map CEVRI outputs to existing policy frameworks and provide concrete examples of how scores could guide decisions like MPA expansions, parametric insurance schemes, or species-specific harvest controls. Without this translational discussion and validation, CEVRI risks remaining an academic exercise rather than a tool for building climate-resilient fisheries.

Methods of analysis: The manuscript lacks empirical validation of the CEVRI framework against observed socio-economic or ecological outcomes, such as historical fishery collapses, livelihood disruptions, or species population declines. Without cross-verification with real-world data (e.g., comparing CEVRI scores to post hoc vulnerability trends), it remains unclear whether these indicators reliably predict risks or offer actionable insights for decision-makers. This omission significantly limits the framework’s credibility for policy or management applications. To address this gap, the authors should either: validate CEVRI scores against historical case studies in the study regions, or compare results with independent vulnerability assessments (e.g., IUCN Red List trends, fishery closure records). At minimum, the discussion must explicitly acknowledge this limitation and outline plans for future validation, as unverified indices risk misdirecting climate adaptation efforts.

Minor comments

Figure 1: The methods should clarify how communities were geographically defined—whether by municipal boundaries, fishing port locations, or dealer reporting addresses. This distinction is critical for interpreting spatial patterns in the analysis. I recommend including a map showing dealer coverage and their respective service areas to help visualize potential data limitations. Additionally, please discuss how landings aggregation might affect the results. For instance, dealers in strategic locations (e.g., Atlantic Beach, FL) may source from multiple states, creating mismatches between reported dealer locations/address and actual fishing grounds. This could artificially inflate or distort the apparent vulnerability of certain communities. Addressing these nuances would strengthen the study’s transparency and highlight important limitations for management applications.

Line 224. Figure 2: For consistency with Figures 3 and 4, please either label the categories directly in Figure 2 (i.e., "1: Low", "2: Medium", "3: High", "4: Very High"), or include this classification system in the figure legend, explicitly defining what each CVA score represents.

Discussion: The CEVRI assessment may be affected by data gaps, particularly regarding small-scale or illegal fisheries activities. Since landings are often attributed to dealer locations rather than actual fishing grounds, this could distort spatial vulnerability patterns. Please discuss how these data limitations might influence the accuracy of CEVRI scores and their interpretation for management purposes.

Discussion: While CEVRI effectively captures exposure and sensitivity components, it currently does not account for community adaptive capacity factors such as alternative livelihoods or policy support systems. I recommend adding a brief discussion acknowledging this limitation and considering how incorporating adaptive capacity metrics might alter vulnerability rankings.

Discussion: The CEVRI framework focuses on climate-specific vulnerabilities without addressing concurrent stressors like overfishing, pollution, and habitat degradation. Please include discussion of how these anthropogenic pressures might interact with or amplify climate impacts, and whether their exclusion might lead to underestimation of true vulnerability in certain ecosystems or communities.

**Do you want your identity to be public for this peer review?** For information about this choice, including consent withdrawal, please see our Privacy Policy

Reviewer #1: **Yes: ** Laura E. Koehn

Reviewer #2: **Yes: ** Eudriano Costa

---

## [Author Response · Author response to Decision Letter 1]

18 Aug 2025

We would like to thank the reviewers for the comments and suggestions that helped improve the manuscript considerably. We have provided detailed responses to each reviewer’s specific comments below. In addition, we have also made improvements to the figures to increase readability and quality.

Reviewer #1: This paper includes an impressive amount of work and includes a lot of great information about risk and vulnerability to fishing communities and fisheries species on the U.S. Northeast and Southeast. It is also very well written and easy to follow. The addition of scores over time is an important, novel contribution. I do however feel there is more work that could be done to strengthen and clarify this manuscript, mainly in the methods.

We thank reviewer 1 for highlighting the novel and important contributions of the paper and for the suggestions that have helped us improve the paper. We give specific responses to each question and suggestions below.

METHODS

1. Use of the term “vulnerability” and no adaptive capacity indicators – the term Total Vulnerability is used but according to line 199, the Total Vulnerability score is only made up of sensitivity and exposure (“considering all 12 sensitivity and 12 exposure factors”) and doesn’t include adaptive capacity. Though the intergovernmental panel on climate change has changed its definition of vulnerability over time, I’m pretty sure in every iteration it includes some form of adaptive capacity. Based on IPCC and other literature, vulnerability is commonly a combination of sensitivity, exposure, and adaptive capacity. Does the Total Vulnerability indicator used include adaptive capacity? If not, why not and should this then be “risk”?

We have made edits to the paper to clarify the use of the terminology referred to above. The indicators were developed to understand potential risk of a community based solely on their level of dependence on certain species and those species bio-environmental vulnerability as calculated by the Climate Vulnerability Assessments conducted for each region by their respective Fisheries Science Center researchers. The term “vulnerability” should only be employed in this context when referring to those prior assessments. This has been corrected in the revised manuscript. It is outside of the scope of this paper to methodologically account for adaptive capacity - this is a very important aspect that we hope to address in a more encompassing future paper but one that would require a great deal of additional effort.

Furthermore, the introduction on lines 144-149 talks about the importance of considering social factors in combination with environmental factors and also adaptive capacity in general. Yet, I don’t think any social adaptive capacity is looked at within the metric of vulnerability used in this manuscript. Total species diversity is looked at as a proxy for adaptive capacity, but what about social components that make up these communities? Yes you found lower risk for the South Atlantic compared to Northeast but what about the variation in social factors between these regions that may make one more able to cope then another? Is it possible to look at how social adaptive capacity factors would impact your results? And/or at least discuss?

Diversity is one aspect widely discussed in the literature as affecting adaptive capacity. The manuscript does not explicitly claim that to be the only factor and this is discussed in the context of literature addressing the relationship between diversity and adaptive capacity. As mentioned above, we believe it is beyond the scope of this paper to include or discuss all aspects that influence a community’s adaptive capacity. Here we focused on the methods used to develop indicators to correlate potential risk and species dependence for communities. We have removed Adaptive Capacity from the keyword list to avoid confusion and have made other edits to increase clarity. In addition we have included a sentence in the discussion to further highlight the value of exploring these relationships in future research:

“Future research to consider factors influencing a community’s adaptive capacity in the face of the environmental variations captured by the CEVRI is also a necessary step for further understanding community vulnerability in a more holistic and pragmatic way.”

2. Lines 182-184 – By using landed value dollars instead of weight, how much does this change what the top species are? Is the high value for certain species potentially due to high value for a rare species and then the top species don’t actually reflect what is most commonly fished for? In other words, would weight actually reflect what is a more common resource across many fishers? For example, Koehn et al. 2022 looked at – “We focused on the top 90% of landed weight, not revenue, to focus on the majority of what is caught in the community (versus a rare but highly valued species caught).”

The reviewer is correct in thinking that the results would be very different if we did the analyses by weight. We made the decision to use value because that better reflects risk from a socio-economic perspective by assessing potential revenue loss for communities, i.e., if environmental changes impact a community’s ability to land species that are of high value that will translate into a significant loss of revenue for the community. Our methods session includes the following clarification which the authors believe is sufficient justification:

“The species’ landed value in dollars was used, instead of weight in pounds, to relate risk to environmental variability to community economic dependence on vulnerable species.”

The idea of high value “rare” species is not one that brings concerns for the regions studied. In the passage quoted above, Koehn et al. were interested in ecological risk rather than socio-economic risk. In their discussion of social vulnerability they also discuss their results in terms of revenue:

“The majority of communities with 25% or more revenue from a single or multiple salmon species are in the top ten percent of at-risk communities (Fig 3, S5 Table). Most communities with greater than or approximately 20% of revenue from Pacific hake or sablefish were also in the top ten percent of at-risk communities.”

3. Lines 219-221 – I know that the Total Sensitivity and Total Vulnerability were taken from another reference, but how were these calculated? How were each of the 12 sensitivity/exposure factors combined and then how was exposure and sensitivity combined? The algorithm for this could change the results.

The references cited for the calculation of the bioenvironmental vulnerability scores (CVAs) have all been peer reviewed and they include details of the methodology and validation strategies used by the authors. We only used the final scores as calculated by the experts in the various studies cited - they have used a well tested methodology to calculate total sensitivity and total vulnerability scores for each species included in the assessment and we have plugged those into the CEVRI formula to calculate the COMMUNITY-level scores for total sensitivity and total vulnerability based on landings composition. We have clarified this in the text by adding the word “community” when referring to the community level total sensitivity and vulnerability.

4. Figure 2 – Flip the y axis so it goes from 0/<0 at the left bottom corner and reads from low to high from up to down. Not sure why the y axis is flipped and goes from high to low

The direction of the values in the y axis is intentional so that the figure reads from left to right in terms of lower to higher risk. We believe that makes the figure more intuitive and it has no impact on the information provided.

5. Lines 238-240 – Why were certain species of importance not included?

We believe this paragraph in the paper (particularly after the inclusion of the underlined below) makes it clear why not all species with importance for communities were included in the CEVRI analyses:

“Although the list of species with CVA scores in the Northeast and Southeast Regions represent a significant portion of the commercial landings in these regions, not all species with regional and local importance were included in the biological assessments, limiting our ability to account for the complete species portfolio for every community. In addition, landings composition varies considerably among different communities in a given region. To ensure that final community scores range between one and four, percent contribution to landings value was calculated based only on the total value of CVA-classified species landed in a community, excluding non-classified species.”

6. Line 272 – The years 2018-2022 were included, did the beginning of covid show any large changes in fishing catch? Was 2020 significantly different than the other years?

This is an interesting question but we only used percent contribution of species to total landings and we did not observe a significant difference in terms of relative species contribution with Covid.

RESULTS

7. Figure 6 has one red community – no red on the legend and what community is this? Why so high for that one community?

This has been fixed.

8. Figures 8-10 – combine into one plot with 3 panels?

Thank you for the suggestion. We will leave this decision to the journal editors.

9. Lines 332-343 and figures 8-10 – What statistics were used to determine that these trends were significant? Could try quantile regression.

We used Pearson’s r - That was indicated in the figure and the caption and we have added it to the text for clarity.

10. I find the community profiles interesting. It does make the paper longer but it’s interesting to draw out these examples. Do most communities fit into similar profiles as the 4 you pulled out?

Each community has a unique profile that is based on their species portfolio. We share the whole list of communities analyzed and their respective CEVRI scores and other variables in the supplemental materials. The criteria used for community selection is clearly explained in the text:

“Four communities representing the different regions and sub-regions studied were selected for profiling. Communities selected displayed a high percentage of landings revenue coming from CVA-classified species and a high value contribution to the regional quotient across the 2018-2022 five-year average (Supplemental Materials I). These profiles exemplify the use of the CEVRI to analyze trends in coastal fishing communities’ level of risk to environmental variability and the relationship between risk levels and landings composition.”

DISCUSSION

11. Lines 504-505 – You mention how social vulnerability and ocean acidification has been discussed elsewhere. What did that paper find? How would that impact your results given there are probably differences in social vulnerability across the regions you looked at?

The reference to the Ekstrom et al (2015) paper - which I believe is what the reviewer is referring to - is used to emphasize the relationship we establish in the discussion about community high dependence on species of shellfish which are highly vulnerable to OA. Similarly to our argument, this previous study reached similar conclusions using a different methodology - that communities highly dependent on highly vulnerable species have more to lose as ocean pH decreases. Other factors discussed would not impact our results because we cannot make inferences about factors the CEVRI is not assessing. We acknowledge that there are many factors that affect a community’s social vulnerability but here we are only discussing the factors that correspond to their revenue dependence on species that are at risk to the environmental factors considered in the CVA. We believe these limitations are made clear in our description of the methods used to calculate the indicators.

12. Lines 518-520 – This is a good point that there may be other environmental factors not considered here that species may be vulnerable to. Have other papers considered these? Also, you note in the methods that Colburn et al. 2016 looked at similar regions – what were their findings compared to yours?

There are many studies that have considered species vulnerability to a number of different environmental factors, including climate change. However, in this study we are using the scores that were calculated as part of the NOAA Fisheries CVA effort to develop the new indicators presented. It is outside of the scope of this paper to consider all possible factors. The CEVRI can be upgraded in future iterations and can be used in conjunction with other indicators and data to show a more complete picture of community risk and vulnerability - we include this in the discussion:

“The community environmental variability risk indicators can also be cross-referenced with other community-level indicators to identify intersections between environmental fluctuations and other factors, such as social and economic vulnerability (NOAA 2024d).”

We do not believe that including comparisons between the current analysis and Colburn et al. (2016) will add anything of value to the paper. That paper presents a coarser version of the analysis done in this manuscript and it is published in a peer reviewed journal so the information is easily accessible.

13. Why do you think the one region’s species diversity vs. vulnerability trend was opposite from the other regions and opposite from what’s expected?

The primary reason for that is that the region in question has high dependence on fewer species that happen to have low CVA scores for Total Vulnerability. This is addressed in the discussion:

“Our analyses also show that the relationship between risk and diversity of landings composition is not always straightforward. Communities in the Gulf of America/Florida Keys sub-region with lower diversity scores also tended to present lower environmental variability risk levels for the factors analyzed. Similarly, the Gulf of America profiled community, Bayou La Batre, AL, scored relatively low in terms of environmental variability risk despite being highly dependent on a small number of shrimp species. While the species in question did not show high vulnerability to the factors analyzed, they may be vulnerable to other environmental fluctuations and risks not considered in our analyses.”

Reviewer #2: Review report

This manuscript entitled “Developing environmental variability risk indicators for fishing communities” presents the development and application of Community Environmental Variability Risk Indicators (CEVRI) to evaluate the vulnerability of U.S. Northeast and Southeast fishing communities to environmental variability. Using commercial landings data (2000–2022) and Climate Vulnerability Assessment (CVA) scores for species, the study aims to assess community-level risk based on dependency on vulnerable species. It is an important and timely contribution, particularly in the context of climate adaptation planning for coastal communities. However, the manuscript suffers from critical methodological, conceptual, and analytical limitations that must be addressed before it can be considered for publication. These include insufficient theoretical framing of “risk,” absence of validation or statistical testing, over-simplified treatment of diversity and adaptive capacity, and vague policy relevance. Please, see my detailed comments below.

We thank the reviewer for the thorough review of the manuscript and the many suggestions that have helped us to improve it. We provide responses to the specific comments and suggestions below.

Major comments

Introduction: The manuscript lacks a formal, operational definition of 'risk'. The distinction between risk, vulnerability, exposure, and adaptive capacity is not clearly articulated. Please, define 'risk' explicitly based on existing frameworks (e.g., IPCC), and clarify how CEVRI maps onto these components.

We believe that what the reviewer means is a better conceptual definition of risk - the operationalization is provided in the methods and that refers to the indicator developed to assess/measure risk. We have made several edits to the i

---

## [Decision Letter · Decision Letter 1]

6 Oct 2025

Developing indicators of risk to environmental variability based on species dependency in U.S. fishing communities in the Northeast and Southeast Regions

PONE-D-25-16131R1

Dear Dr. Tarsila Seara,

We’re pleased to inform you that your manuscript has been judged scientifically suitable for publication and will be formally accepted for publication once it meets all outstanding technical requirements.

Kind regards,

Tzen-Yuh Chiang

Academic Editor

PLOS ONE

Additional Editor Comments (optional):

Reviewers' comments:

Reviewer's Responses to Questions

**Comments to the Author**

Reviewer #1: All comments have been addressed

Reviewer #2: All comments have been addressed

2. Is the manuscript technically sound, and do the data support the conclusions?

Reviewer #1: Yes

Reviewer #2: Yes

3. Has the statistical analysis been performed appropriately and rigorously?

Reviewer #1: Yes

Reviewer #2: Yes

4. Have the authors made all data underlying the findings in their manuscript fully available?

Reviewer #1: Yes

Reviewer #2: No

5. Is the manuscript presented in an intelligible fashion and written in standard English?

Reviewer #1: Yes

Reviewer #2: Yes

Reviewer #1: The authors have addressed all my previous comments. My last comments are more related to improving the readability of the paper than anything with the analysis. Specifically, there first two paragraphs of the introduction are extensively long on the temperature and ocean acidification changes in marine environments in general. These could be reduced to one shorter overall paragraph on climate change in the ocean before diving into the specifics of the regions the authors are looking at. Most readers know of overall changes in the marine environment and are more interested in the specifics of the regions at hand. Finally, there are a multitude of figures in this paper. Some figures are repetitive but for different regions and could be put into a multi-panel 1 figure. For instance, figures 8-10 could be a 3 panel single figure. This would then make it easier to compare results across the regions. I understand why map figures are single figures but for other types of figures, these could be combined into multi-panel single figures.

Reviewer #2: The authors have substantially improved the manuscript and have addressed all my main questions. I am satisfied with their responses and therefore endorse the manuscript for publication.

**Do you want your identity to be public for this peer review?** For information about this choice, including consent withdrawal, please see our Privacy Policy

Reviewer #1: No

Reviewer #2: No

---

## [Editor Report · Acceptance letter]

PONE-D-25-16131R1

PLOS ONE

Dear Dr. Seara,

I'm pleased to inform you that your manuscript has been deemed suitable for publication in PLOS ONE. Congratulations! Your manuscript is now being handed over to our production team.

Kind regards,

on behalf of

Dr. Tzen-Yuh Chiang

Academic Editor

PLOS ONE